# Aberrant cell segregation in the craniofacial primordium and the emergence of facial dysmorphology in craniofrontonasal syndrome

**Terren K. Niethamer**[1,2,3,4☯‡], **Teng Teng**[1,2,3☯‡], **Melanie Franco**[1,2,3], **Yu Xin Du**[1,2,3], **Christopher J. Percival**[5]*, **Jeffrey O. Bush**[1,2,3,4]*

**1** Program in Craniofacial Biology, University of California San Francisco, San Francisco, California, United States of America, **2** Department of Cell and Tissue Biology, University of California San Francisco, San Francisco, California, United States of America, **3** Institute for Human Genetics, University of California San Francisco, San Francisco, California, United States of America, **4** Biomedical Sciences Graduate Program, University of California San Francisco, San Francisco, California, United States of America, **5** Department of Anthropology, Stony Brook University, Stony Brook, New York, United States of America

☯ These authors contributed equally to this work.
‡ These authors share first authorship on this work.
* Christopher.Percival@stonybrook.edu (CJP); Jeffrey.Bush@ucsf.edu (JOB)

**Data Availability Statement:** All relevant data are within the manuscript and its Supporting Information files.

## Abstract

Craniofrontonasal syndrome (CFNS) is a rare X-linked disorder characterized by craniofacial, skeletal, and neurological anomalies and is caused by mutations in *EFNB1*. Heterozygous females are more severely affected by CFNS than hemizygous males, a phenomenon called cellular interference that results from EPHRIN-B1 mosaicism. In *Efnb1* heterozygous mice, mosaicism for EPHRIN-B1 results in cell sorting and more severe phenotypes than *Efnb1* hemizygous males, but how craniofacial dysmorphology arises from cell segregation is unknown and CFNS etiology therefore remains poorly understood. Here, we couple geometric morphometric techniques with temporal and spatial interrogation of embryonic cell segregation in mouse mutant models to elucidate mechanisms underlying CFNS pathogenesis. By generating EPHRIN-B1 mosaicism at different developmental timepoints and in specific cell populations, we find that EPHRIN-B1 regulates cell segregation independently in early neural development and later in craniofacial development, correlating with the emergence of quantitative differences in face shape. Whereas specific craniofacial shape changes are qualitatively similar in *Efnb1* heterozygous and hemizygous mutant embryos, heterozygous embryos are quantitatively more severely affected, indicating that *Efnb1* mosaicism exacerbates loss of function phenotypes rather than having a neomorphic effect. Notably, neural tissue-specific disruption of *Efnb1* does not appear to contribute to CFNS craniofacial dysmorphology, but its disruption within neural crest cell-derived mesenchyme results in phenotypes very similar to widespread loss. EPHRIN-B1 can bind and signal with EPHB1, EPHB2, and EPHB3 receptor tyrosine kinases, but the signaling partner(s) relevant to CFNS are unknown. Geometric morphometric analysis of an allelic series of *Ephb1; Ephb2; Ephb3* mutant embryos indicates that EPHB2 and EPHB3 are key receptors mediating *Efnb1* hemizygous-like phenotypes, but the complete loss of EPHB1-3 does not fully

**Funding:** Sources of funding include R01DE023337 from NIH/NIDCR to J.O.B. and F31DE026059 from NIH/NIDCR to T.K.N. The funders had no role in study design, data collection and analysis, decision to publish, or preparation of the manuscript.

**Competing interests:** The authors have declared that no competing interests exist.

recapitulate the severity of CFNS-like *Efnb1* heterozygosity. Finally, by generating *Efnb1$^{+/\Delta}$; Ephb1; Ephb2; Ephb3* quadruple knockout mice, we determine how modulating cumulative receptor activity influences cell segregation in craniofacial development and find that while EPHB2 and EPHB3 play an important role in craniofacial cell segregation, EPHB1 is more important for cell segregation in the brain; surprisingly, complete loss of EPHB1-EPHB3 does not completely abrogate cell segregation. Together, these data advance our understanding of the etiology and signaling interactions underlying CFNS dysmorphology.

## Author summary

Craniofacial anomalies are extremely common, accounting for one third of all birth defects, but even when the responsible genes are known, it often remains to be determined exactly how development has gone wrong. Craniofrontonasal syndrome (CFNS), which affects multiple aspects of craniofacial development, is a particularly mysterious disorder because it is X-linked, but affects females more severely than males, the opposite situation of most X-linked diseases. The responsible gene has been identified as *EFNB1*, which encodes the EPHRIN-B1 signaling molecule. Why *EFNB1$^{+/-}$* heterozygous females exhibit severe stereotypical CFNS phenotypes is not well understood, but it is related to the fact that X chromosome inactivation generates mosaicism for EPHRIN-B1 expression. Using mice harboring mutations in the *Efnb1* gene in different embryonic tissues, and in receptor genes *Ephb1-3*, together with quantitative methods to measure craniofacial structures in developing embryos, we establish the tissue-specific contributions of EPHRIN-B1 mosaicism to craniofacial dysmorphology. We also examine when EPHRIN-B1 regulates cellular position during different stages of craniofacial development and which EPHB receptors are involved. Our results reveal the specific cellular context and signaling interactions that are likely to underlie CFNS and provide new understanding of how EPHRIN-B1 may regulate normal craniofacial development.

## Introduction

Congenital craniofacial anomalies account for one third of all birth defects [1]. Advances in craniofacial genetics have identified many genes involved in craniofacial syndromes [2], but an understanding of the underlying etiology and progression over developmental time for each condition will be necessary for improved therapies for this large group of disorders. Craniofrontonasal syndrome (CFNS, OMIM #304110) is a form of frontonasal dysplasia that is caused by loss of function mutations in the *EFNB1* gene, which is located on the X chromosome [3–5]. Paradoxically, although this syndrome is X-linked, *EFNB1* heterozygous females are severely affected by CFNS, whereas males with hemizygous loss of *EFNB1* function appear mildly affected or unaffected; this phenomenon is termed "cellular interference," though how this difference in severity arises is currently unknown [4–6]. Heterozygous female patients frequently display a combination of orbital hypertelorism based on measurements of inner canthal and interpupillary distances or on computed tomography (CT) scans, a short and wide upper face, facial asymmetry, unilateral or bilateral coronal craniosynostosis, a short nose, bifid nasal tip, and a broad nasal bridge [3–5,7]. In a subset of cases, cleft lip and palate, agenesis of the corpus callosum [4], and maxillary hypoplasia [7] have also been noted. In addition to craniofacial defects, patients present with skeletal defects including syndactyly and polydactyly.

CFNS has been termed a neurocristopathy, and it has been hypothesized that CFNS phenotypes may be partly attributable to impacts on early neural crest cell (NCC) migration or to later bone differentiation defects [4,8–10]; however, the developmental etiology of this disorder remains unknown. Because CFNS patients are clinically evaluated postnatally but craniofacial development begins very early during embryogenesis, it is difficult to pinpoint the developmental timing and tissue origin of the craniofacial phenotypes. Hypertelorism, frontonasal dysplasia and widened midface are key defining phenotypes that may have a variety of embryologic tissue origins. It is possible that these changes are due to early defects in NCCs, but they could also be secondary to changes in morphology of the brain and/or neurocranium, or could be caused by later changes in the morphogenesis of craniofacial structures. The forebrain develops in close interaction with the developing midface, and provides a physical substrate that shapes the midface [11,12]. Reduced brain growth correlates with reduced facial growth in a short-faced mutant mouse model [13], and in humans, brain shape differences were found to be correlated with the occurrence of cleft lip with or without cleft palate (CL/P) and cleft palate only (CPO) [14]. Increases in brain size could underlie clefting phenotypes by increasing separation of the facial prominences to an extent that they can no longer make contact, even if their outgrowth is normal [15,16]. Molecular signaling from the brain to the developing midface can also impact craniofacial morphogenesis and contributes to hypotelorism, and possibly hypertelorism [17–20]. Facial dysmorphology may also be secondary to other skull phenotypes, including craniosynostosis, which restricts the directions of skull growth [21,22] or to modified cranial base growth [23,24]. However, evidence of effects of craniosynostosis syndrome mutations on early facial shape highlight that frontonasal dysplasia can also be a primary result of local developmental perturbations of facial prominence growth patterns [25–27].

*Efnb1* encodes EPHRIN-B1, a member of the Eph/ephrin family of membrane-linked signaling molecules; signaling between EPH receptors and EPHRINs is important for boundary formation, cell migration, axon guidance, vascular development, and neurogenesis [28–36]. Analysis of several tissue types indicates that X-inactivation is not biased by *EFNB1* mutation [4,37], suggesting that loss of gene function does not broadly impact cell survival or competition. Instead, mosaicism for EPHRIN-B1 expression results in more severe dysmorphogenesis, as rare male patients with severe CFNS phenotypes exhibit somatic mosaicism for *EFNB1* mutations [37–39]. Mosaicism for *Efnb1* mutation has been demonstrated to result in cell segregation between EPHRIN-B1 expressing and non-expressing cells in mice [40–42], though the timing of onset and tissue origin of segregation relevant to CFNS was not established in these studies. More recently, we have demonstrated that cell segregation occurs in the early neural plate in $Efnb1^{+/-}$ mouse embryos, and in neuroectodermal cells differentiated from CFNS patient iPSCs [43,44], but it is unknown whether this cell segregation contributes to craniofacial phenotypes.

As in human CFNS, mosaic loss of EPHRIN-B1 expression in $Efnb1^{+/-}$ mice leads to additional phenotypes not found in hemizygous ($Efnb1^{-/Y}$) or homozygous ($Efnb1^{-/-}$) loss in mice [8,41,42]. Although this mouse model is considered to phenocopy CFNS, the facial forms of heterozygous and hemizygous mice have not been described beyond the report of relatively high frequency of cleft palate and shorter skulls [8,9,42]. In addition, the relationship between timing and tissue specificity of cell segregation and phenotypic progression of CFNS craniofacial phenotypes is unknown, and how EPHRIN-B1-mediated segregation contributes to facial dysmorphogenesis therefore remains mysterious.

Here, we use mouse models of CFNS to determine the timing and cell type specificity of EPHRIN-B1-mediated cell segregation as it relates to the onset and progression of craniofacial phenotypes. We compare the facial form of *Efnb1* heterozygous female and hemizygous male embryos with control embryos across four stages of craniofacial development to quantify the

specific effects of *Efnb1* loss on facial growth and development to better understand the ontogeny of CFNS dysmorphology. Through tissue-specific generation of *Efnb1* mosaicism, we demonstrate that EPHRIN-B1 is a potent regulator of cell segregation in multiple cell types across craniofacial development and that the timing of segregation in craniofacial primordia correlates with the onset and progression of facial phenotypes in developing embryos. Next, through morphometric analysis of an allelic series of compound *Ephb1; Ephb2; Ephb3* receptor gene mutants, we assess the relative contributions of each receptor to craniofacial morphogenesis. Finally, by generating *Efnb1*^{+/Δ} embryos with combinatorial compound loss of receptors, we determine the likely EPHRIN-B1 signaling partners that drive CFNS cell segregation. Together, these results indicate that cell segregation occurring in post-migratory mesenchymal populations of the craniofacial primordia is facilitated by numerous EPHRIN-B1 receptors and is likely the principal driver of cellular interference and severe facial dysmorphogenesis in CFNS.

## Results

### *Efnb1* mutant genotype has a significant effect on embryonic facial shape from E11.5 to E14.5 that mirrors CFNS

Robust quantitative methods are required to investigate the effects of mosaic expression of EPHRIN-B1 on facial morphology. To compare phenotypic severity between heterozygous females and hemizygous males over time, we quantified mouse embryo facial shape at progressive daily stages between E11.5 and E14.5 using geometric morphometrics analysis of landmarks collected on micro-computed tomography (μCT) derived facial surfaces of *Efnb1*^{+/Δ} and *Efnb1*^{Δ/Y} embryos as well as a pooled control sample of *Efnb1*^{+/lox} and *Efnb1*^{lox/Y} embryos that we refer to as *Efnb1*^{wt}. To determine the significance and relative contribution of facial size (estimated as centroid size) and *Efnb1* genotype in determining facial shape, we carried out a Procrustes ANOVA analysis on E11.5 embryos using a published landmark set [45]. Facial size and *Efnb1* genotype both contribute significantly to facial shape of E11.5 embryos (Table 1), explaining approximately 23% and 11% of the facial shape variation, respectively. The significant genotype effect indicates that EPHRIN-B1 mosaicism or loss influences facial

**Table 1. Significant influences on facial shape at E11.5 (Procrustes ANOVA).**

|  | Df | SS | MS | Rsq | F | Z | Pr($>$F) |
|---|---|---|---|---|---|---|---|
| **Size**[a] | 1 | 0.141 | 0.141 | 0.229 | 24.719 | 6.274 | 0.001[*] |
| **Genotype**[b] | 2 | 0.069 | 0.034 | 0.111 | 6.005 | 5.970 | 0.001[*] |
| **Residuals** | 71 | 0.406 | 0.006 | | | | |
| **Total** | 74 | 0.615 | | | | | |

[a]Estimate of the influence of overall size (estimated as centroid size) on facial shape.

[b]Estimate of the influence of genotype (as a factor) on facial shape.

Df is the degrees of freedom for each factor.

SS is the total sum of the squares for each factor, based on Procrustes distances.

MS is the mean sum of squares (i.e. SS/Df) for each factor.

Rsq provides an estimate of how much facial shape variance a given factor explains.

F is a test statistic comparing MS of a factor to the amount of variation within factor groups. A high F statistic suggests that the factor has a significant effect on facial shape variation.

Z is an effect size associated with the F statistic.

Pr($>$F) is the probability that you would get an F score higher than this factor's F score by chance.

[*]indicates a significant effect on facial shape (α = 0.05), as calculated using a permutation test.

shape as early as E11.5 in mice. We interrogated genotype-specific effects on facial shape to pinpoint regions where differences occur. Landmark-specific shape change vectors for both $Efnb1^{\Delta/Y}$ (n = 21) and $Efnb1^{+/\Delta}$ (n = 26) mutant genotypes indicate increased facial width and decreased facial height, with maxillary prominences more posterior in relation to neurocranial landmarks compared with $Efnb1^{wt}$ (n = 28) (S1 Fig). Overall, there is evidence of reduced anterior outgrowth of and greater lateral distance between the facial prominences in $Efnb1$ mutant embryos.

Given a significant effect of the $Efnb1$ genotype on facial shape at E11.5, we performed morphometric analysis on E12.5-E14.5 embryos to determine whether there was a change in the severity or type of facial dysmorphology as the face grows. We used a novel landmark set that better captures facial shape at these specific stages (S2 Fig). A Procrustes ANOVA analysis with facial size (estimated as centroid size), embryonic age, and $Efnb1$ genotype as factors indicated that each contributes significantly to facial shape (Table 2). Additionally, the interaction between age and genotype has a significant effect on facial shape. As expected for a sample covering multiple embryonic days, facial shape variation correlated with size (i.e., allometry) explained 77% of facial shape variation. The significant effect of $Efnb1$ genotype explained almost 7% of facial shape variation. Visualization of landmark vectors illustrating genotype-specific shape effects indicate overall similarities in the effects of $Efnb1^{\Delta/Y}$ (n = 25) and $Efnb1^{+/\Delta}$ (n = 29) genotypes on facial shape at E14.5 (Fig 1A–1H). Both mutant genotypes display hypertelorism, represented by an increased relative width between anterior eye landmarks. They also have a relatively inferior-posterior nose, anterior ear, and latero-posterior lip corners. Whereas $Efnb1^{\Delta/Y}$ embryos exhibited shorter faces, the degree of facial shortening was more extreme in $Efnb1^{+/\Delta}$ embryos, as demonstrated by longer vectors at the ear and nose landmarks (Fig 1H). Together, these shared patterns of dysmorphology indicate that hypertelorism and facial shortening occur in both male hemizygotes and female heterozygotes.

Similarities between E12.5-E14.5 and E11.5 mutant genotype effects suggest a continuity of shape dysmorphology between E11.5 and E14.5. However, it was important to verify that effects at different embryonic ages remain parallel after accounting for normal facial growth across this developmental period. Given that 77% of facial variation of the E12.5-E14.5 sample was explained by size, it was not surprising that the first principal component (PC) of a principal component analysis (PCA) of facial shape separates specimens in this sample by embryonic age (Fig 1I). A multivariate linear model was used to estimate the allometric component of shape variation that is common across the sample regardless of genotype (Fig 1J). The residuals

**Table 2. Significant influences on facial shape from E12.5-E14.5 (Procrustes ANOVA).**

|  | Df | SS | MS | Rsq | F | Z | Pr(>F) |
|---|---|---|---|---|---|---|---|
| **Size**[a] | 1 | 1.706 | 1.706 | 0.772 | 1083.475 | 7.158 | 0.001* |
| **Genotype**[b] | 2 | 0.145 | 0.072 | 0.066 | 46.005 | 13.728 | 0.001* |
| **Age**[c] | 1 | 0.011 | 0.011 | 0.005 | 7.287 | 9.243 | 0.001* |
| **Genotype:Age**[d] | 2 | 0.016 | 0.008 | 0.007 | 5.207 | 10.914 | 0.001* |
| **Residuals** | 210 | 0.331 | 0.002 |  |  |  |  |
| **Total** | 216 | 2.210 |  |  |  |  |  |

[a]Estimate of the influence of overall size (estimated as centroid size) on facial shape.

[b]Estimate of the influence of genotype (as a factor) on facial shape.

[c]Estimate of the influence of age (as continuous) on facial shape across E12.5-E14.5 specimens.

[d]Genotype:Age is the interaction effect of genotype and age.

Column abbreviations are the same as defined for Table 1.

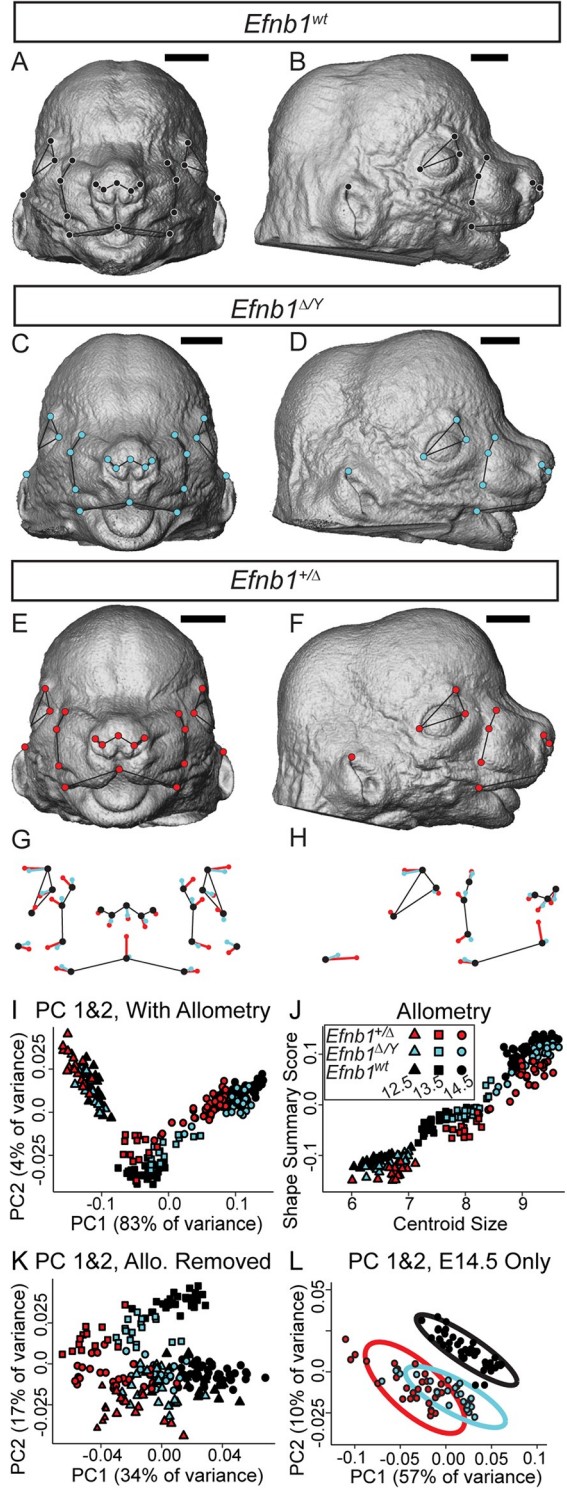

**Fig 1.** *Efnb1* **mutant embryos have quantitative facial shape effects that mimic CFNS. (A-F)** Facial landmarks identified on representative *Efnb1^wt* **(A-B)**, *Efnb1^{Δ/Y}* **(C-D)**, and *Efnb1^{+/Δ}* **(E-F)** E14.5 specimen surfaces. *Scale bar, 1000 μm* **(G-H)** Common facial shape effects of *Efnb1^{Δ/Y}* (cyan) and *Efnb1^{+/Δ}* (red) genotypes on facial landmark position, compared to *Efnb1^wt* (black) from the **(G)** anterior and **(H)** lateral views. The lengths of these shape difference vectors are magnified three times to allow for easy comparison. Thin black lines are placed for anatomical reference. **(I-L)** Plots to illustrate facial shape variation of *Efnb1^{Δ/Y}* (cyan) and *Efnb1^{+/Δ}* (red) and *Efnb1^wt* (black) genotypes across E12.5 (triangle), E13.5 (square), and E14.5 (circle). **(I)** Facial shape variation across E12.5–14.5

specimens is illustrated along the first two principal components. **(J)** A linear relationship exists between facial size and a multivariate summary score of facial shape, which indicates a strong allometric effect across this period of development. **(K)** The first two principal components of facial shape after accounting for this developmental allometry illustrate a common genotype effect across ages. **(L)** Facial shape variation of only E14.5 specimens, with 95% confidence intervals, illustrates similarities in the effect of both genotypes compared to control specimens. Number of embryos analyzed is presented in S1 Table.

of this regression are interpreted as facial shape after accounting for size related shape variation. The first PC of a PCA of these facial shape residuals represents a common axis of facial shape covariation that separates genotypes (Fig 1K), suggesting major similarities in mutant genotype effects on facial shape across embryonic ages. Although individual PCs illustrate patterns of facial shape covariation, they each represent only part of overall covariation. Therefore, we calculated Procrustes distances between mean control and affected genotype facial shapes to confirm the significance of mean facial shape differences between genotypes and to estimate the relative severity of facial shape dysmorphology. There were significant differences in mean facial shape between control and each mutant genotype at all embryonic ages (Table 3). In addition, within each age, the mean facial shapes of $Efnb1^{+/\Delta}$ embryos were always more different from $Efnb1^{wt}$ controls than were $Efnb1^{\Delta/Y}$ facial shapes. After accounting for normal growth processes, the genotypes separate along the same axis at E12.5, E13.5, and E14.5, indicating strong similarities in genotype across ages. The increasing Procrustes distances between wildtype and the mean shapes of both mutant genotypes between E12.5 and E14.5 indicates increasing severity of a qualitatively similar dysmorphology across this period of growth. Based on our analysis, $Efnb1^{+/\Delta}$ and $Efnb1^{\Delta/Y}$ embryos display similar types of dysmorphology that increases in severity between E12.5 and E14.5, with $Efnb1^{+/\Delta}$ females displaying quantitatively greater severity than $Efnb1^{\Delta/Y}$.

While the overall phenotypes are generally similar across ages and between $Efnb1$ mutant genotypes, there are some specific differences in $Efnb1^{+/\Delta}$ and $Efnb1^{\Delta/Y}$ genotype effects (Fig 1A–1H). For example, $Efnb1^{+/\Delta}$ embryos display increased relative width of the posterior whisker margins and a posterior-inferior corner of the whisker region, whereas $Efnb1^{\Delta/Y}$ embryos do not. This suggests a larger increase in relative width of the midfacial region in the female heterozygotes compared to male hemizygotes. In addition, $Efnb1^{+/\Delta}$ embryos display a reduced length of the midline connection between the whisker pads, that appeared as a midline notch in the upper lip, possibly analogous to the shortened human filtrum described in CFNS (Fig 1A,

**Table 3. Age-specific comparisons of the Procrustes distances between the mean shape of affected and control genotypes, after accounting for allometry.**

| | | *Efnb1* genotype | |
|---|---|---|---|
| | wildtype (95% CI) | $\Delta/Y$[a] | $+/\Delta$[a] |
| E11.5^ | 0.07–0.18^ | 0.22*^ | 0.32*^ |
| E12.5 | 0.04–0.09 | 0.15* | 0.23* |
| E13.5 | 0.03–0.06 | 0.19* | 0.28* |
| E14.5 | 0.03–0.06 | 0.18* | 0.29* |

[a] Higher values represent a greater difference in facial shape, a proxy for severity of dysmorphology.

* indicates a significantly different facial shape than control, based on the 95% control confidence intervals produced by bootstrapping the control sample.

^ indicates that E11.5 Procrustes distance values cannot be directly compared to E12.5-E14.5 values, because they are based on a different landmark set and separate Procrustes superimposition. However, the pattern of the ordering of Procrustes distance values within ages can be compared and show similar patterns of significance.

1C, 1E and 1G) [46]. Overall, however, our results demonstrate that increased midfacial expansion is exacerbated in *Efnb1*$^{+/Δ}$ embryos compared with *Efnb1*$^{Δ/Y}$ embryos, rather than resulting from distinct effects on additional craniofacial structures.

## EPHRIN-B1-mediated cell segregation occurs in post-migratory neural crest-derived craniofacial mesenchyme

Cell segregation has been proposed to underlie increased severity in heterozygous female CFNS patients with EPHRIN-B1 mosaicism. We have previously shown that cell segregation first occurs in the headfold of E8.5 *Efnb1*$^{+/Δ}$ embryos prior to NCC emigration [44], suggesting the possibility that early segregation of NCC progenitors might result in the cellular distribution patterns we observe at later stages. Alternatively, later segregation within post-migratory NCC-derived populations could result in increased CFNS severity. *Sox10* is expressed throughout NCCs prior to their emigration, and in *Sox10-Cre*$^{Tg/0}$; *ROSA26*$^{mTmG/+}$ reporter embryos, we observed robust recombination throughout the post-migratory NCCs, including the maxillary process (MXP) and the frontonasal prominence (FNP) (S3A and S3B Fig). To determine when and where cell segregation was occurring, we utilized a ubiquitously expressed X-linked GFP (XGFP) transgenic allele to monitor normal patterns of X chromosome inactivation (XCI) compared with EPHRIN-B1 expression as detected by immunofluorescence (IF) at distinct stages of development [44,47,48]. We generated NCC-specific EPHRIN-B1 mosaic *Efnb1*$^{+XGFP/lox}$; *Sox10-Cre*$^{Tg/0}$ embryos and examined them for segregation at E10.5, after NCCs have populated the craniofacial mesenchyme, quantifying cell segregation by measuring the size of XGFP expressing cell patches. In controls, patches of XGFP cells are small and intermixed, whereas we expect cell sorting to result in fewer patches that contain a greater number of cells. Notably, *Efnb1*$^{+XGFP/lox}$; *Sox10-Cre*$^{Tg/0}$ embryos did not exhibit cell segregation in the MXP at E10.5 (S3G and S3I Fig; **n = 6**) and instead resembled control *Efnb1*$^{+XGFP/lox}$ embryos (S3C, S3D, S3I and S3J Fig; **n = 4**), indicating that cell segregation in premigratory and migratory NCCs, if it occurs, does not carry through to give rise to segregated populations in post-migratory NCC-derived MXP mesenchyme. Consistent with the absence of segregation in the MXP of both *Efnb1*$^{+XGFP/lox}$; *Sox10-Cre*$^{Tg/0}$ and *Efnb1*$^{+XGFP/Δ}$ embryos (S3E and S3G Fig), we found that EPHRIN-B1 expression was low in the MXP at this stage (S3C Fig; **n = 4**). EPHRIN-B1 expression was higher in the FNP at E10.5 (S3D Fig; **n = 4**), which correlated with a small, but statistically significant increase in cell segregation in the FNP of both *Efnb1*$^{+XGFP/lox}$; *Sox10-Cre*$^{Tg/0}$ (S3H and S3J Fig; **n = 6**) and *Efnb1*$^{+XGFP/Δ}$ embryos (S3F and S3J Fig; **n = 3**) at this stage. However, whereas E11.5 control *Efnb1*$^{+XGFP/lox}$ embryos exhibited a fine-grained mosaic pattern of XGFP expression in the MXP and FNP (Fig 2A and 2B; S3I Fig; **n = 4**), in *Efnb1*$^{+XGFP/lox}$; *Sox10-Cre*$^{Tg/0}$ NCC mosaic embryos, distinct large segregated patches of XGFP expression were more abundant, and fewer small patches of individual XGFP cells were observed in both structures (Fig 2C and 2D; S3I and S3J Fig; **n = 4**), indicating that EPHRIN-B1 drives segregation in the post-migratory NCC-derived mesenchyme.

## Post-migratory neural crest cell segregation results in local dysmorphogenesis of craniofacial structures

The finding that segregation occurs in E11.5 craniofacial mesenchyme demonstrates that EPHRIN-B1 mediates this process after NCC migration is completed. We next wished to determine whether segregation continues into later stages of craniofacial development. EPHRIN-B1 has strong expression in the anterior secondary palate mesenchyme, and loss of function of *EFNB1* may result in cleft palate in both humans and mice [3,4,41,49,50]. We therefore asked whether palatal mesenchyme cells mosaic for EPHRIN-B1 expression can

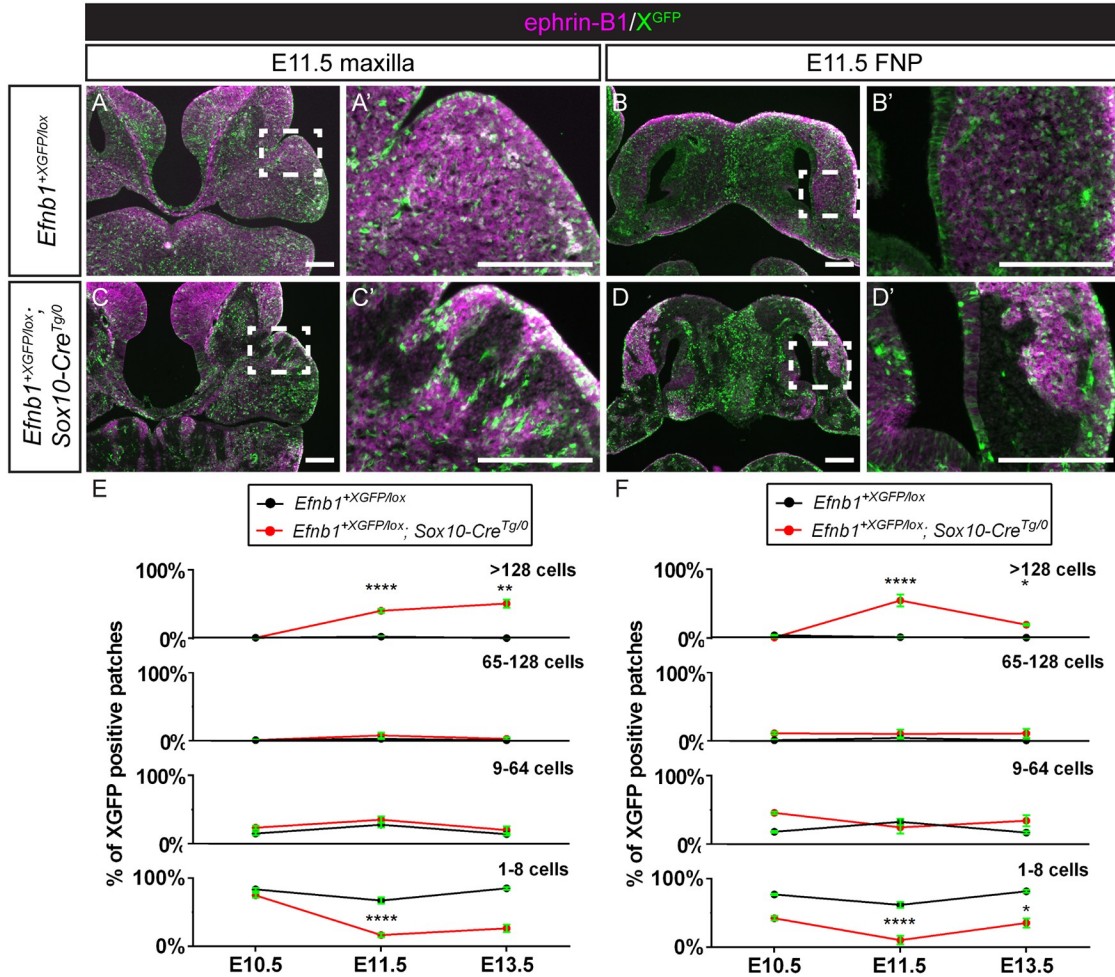

**Fig 2. Post-migratory neural crest cells mosaic for EPHRIN-B1 expression undergo cell segregation in craniofacial primordia.**
**(A, A')** Immunostaining E11.5 frontal sections for EPHRIN-B1 (magenta) and GFP (green) reveals that *Efnb1*$^{+XGFP/lox}$ control embryos demonstrate a fine-grained mosaic pattern of XGFP expression, and EPHRIN-B1 expression is strong in the maxillary prominences and **(B, B')** the lateral FNP. **(C, C')** *Efnb1*$^{+XGFP/Δ}$; *Sox10-Cre*$^{Tg/0}$ embryos with EPHRIN-B1 mosaicism specifically in NCCs show dramatic cell segregation in the maxillary prominences and **(D, D')** the lateral FNP, indicating that NCCs are capable of undergoing EPHRIN-B1-mediated segregation resulting in aberrant EPHRIN-B1 expression patterns in craniofacial mesenchyme. *Scale bars, 200 μm.* **(E)** Distribution of percentage of XGFP positive patches of various sizes over time in the MXP/secondary palate. Means of the size distributions across all sections measured for a given genotype are plotted, error bars represent S.E.M., **P<0.01; ****, P<0.0001 for comparison of each timepoint with the preceding timepoint **(F)** Distribution of percentage of XGFP positive patches of various sizes over time in the FNP. Means of the size distributions across all sections measured for a given genotype are plotted, error bars represent S.E.M., *, P<0.05; ****, P<0.0001 for comparison of each timepoint with the preceding timepoint. Number of embryos analyzed is presented in S1 Table.

undergo segregation by utilizing the *Shox2*$^{IresCre}$ mouse line, as *Shox2* is expressed in a similar domain to EPHRIN-B1 in the anterior secondary palate [41,51,52]. Though *Shox2*$^{IresCre}$ mediated recombination was observed in neurofilament-expressing maxillary trigeminal ganglion nerve cells at E11.5 (S4A and S4B Fig; **n = 2**), recombination in the anterior palatal mesenchyme was first apparent at E12.5 (S4C and S4D Fig; **n = 2**). Consistent with this timing of *Shox2*$^{IresCre}$ onset, we observed no segregation in *Efnb1*$^{+XGFP/lox}$; *Shox2*$^{IresCre/+}$ embryos at E11.5 (S4E and S4F Fig; **n = 3**) but small groups of segregated XGFP expressing cells became apparent in 3/4 E12.5 *Efnb1*$^{+XGFP/lox}$; *Shox2*$^{IresCre/+}$ embryos we examined (S4H Fig; **n = 4**) compared with *Efnb1*$^{+XGFP/lox}$ control embryos (S4G Fig; **n = 3**). EPHRIN-B1 is therefore a

driver of segregation not only in the headfold and NCC progenitor cells, but also in post-migratory craniofacial mesenchyme. These data demonstrate that EPHRIN-B1-mediated cell movements continue through development of craniofacial structures, and segregation within these structures may continually contribute to CFNS dysmorphology.

We have demonstrated that differences in facial shape are evident in female $Efnb1^{+/\Delta}$ heterozygous embryos as early as E11.5, but these shape changes continue to develop over time and increase in severity through E14.5. We next examined how segregation later in development correlates with changes to craniofacial tissue morphology by examining embryos with EPHRIN-B1 mosaicism in specific cell types at E13.5. Control embryos have strong EPHRIN-B1 expression in the tips of the anterior palatal shelves and lateral FNP consistent with the CFNS-like phenotypes we discovered by morphometric analysis, while XGFP is visible in a fine-grained mosaic pattern in each structure (Figs 3A and 4A; **n = 9**). In full $Efnb1^{+-XGFP/\Delta}$ heterozygotes, large EPHRIN-B1/GFP expressing and non-expressing patches correlated with aberrant EPHRIN-B1 expression boundaries, including irregularities of palatal shelf shape (Fig 3B; **n = 5**) and apparent bifurcations of the nasal conchae (Fig 4B; **n = 5**). Neural crest-specific mosaic $Efnb1^{+XGFP/lox}$; $Sox10\text{-}Cre^{Tg/0}$ embryos exhibited a similar correspondence between EPHRIN-B1/XGFP patches and local dysmorphology in both the secondary palatal shelves (Fig 3C **n = 5**) and nasal conchae (Fig 4C; **n = 5**). Interestingly, in palate mesenchyme-specific $Efnb1^{+XGFP/lox}$; $Shox2^{IresCre/+}$ heterozygotes, EPHRIN-B1/XGFP expressing and non-expressing patches were apparent in the E13.5 anterior palate mesenchyme (Fig 3D). These patches appeared somewhat smaller than those in full or NCC-specific mosaic embryos, though quantification of cell segregation did not reveal a statistically significant difference in XGFP patch size compared with $Efnb1^{+XGFP/lox}$ control embryos (Fig 3A, 3D, 3E and 3F). The palatal shelves of $Efnb1^{+XGFP/lox}$; $Shox2^{IresCre/+}$ embryos did not seem as dramatically dysmorphic as $Efnb1^{+XGFP/\Delta}$ heterozygotes, though local bending occurred at EPHRIN-B1 expression boundaries with small bumps surrounding the boundary (Fig 3D, **n = 5**). No segregation was evident in the FNP of palate mesenchyme-specific $Efnb1^{+XGFP/lox}$; $Shox2^{IresCre/+}$ heterozygotes, with no local dysmorphology in the nasal conchae (Fig 4D; **n = 4**). In total, these data demonstrate that EPHRIN-B1 mediates segregation in the post-migratory NCC-derived mesenchyme of two structures key to CFNS pathology and that these boundaries correlate with tissue structure dysmorphology.

### Tissue-specific contributions to CFNS dysmorphology

The expression patterns of EPHRIN-B1 in the early neural plate, telencephalon and post-migratory craniofacial neural crest, together with the finding that cell segregation can occur independently in each of these contexts, led us to ask whether disruption in distinct tissues contributes to CFNS dysmorphology. We have previously shown that EPHRIN-B1 mediates segregation in the neural plate neuroepithelium and that segregation is apparent in the developing brain [44,53]. In a mouse model of a different neurocristopathy, Treacher Collins syndrome, apoptosis of neuroepithelial cells is observed together with a reduction in cranial NCCs [54,55]. In addition, changes in brain shape can indirectly cause changes to facial shape [11,12]. We therefore wondered whether EPHRIN-B1 mosaicism in the brain could result in changes to facial shape. Sox1$^{Cre}$ mediates recombination in the neural plate as early as E8.5 [56], and crossing to the $ROSA26^{mTmG}$ reporter revealed widespread recombination throughout the brain at E13.5 (S5A Fig; **n = 6**) and with varying recombination in relatively few sparse mesenchymal cells in craniofacial structures such as the palatal shelves and FNP (S5B, S5C–S5F' Fig; **n = 6**). Compared with control embryos (Fig 5A; **n = 9**), $Efnb1^{+XGFP/lox}$; $Sox1^{Cre/+}$ embryos exhibited robust segregation in the telencephalon of the brain (Fig 5C, 5D and 5E;

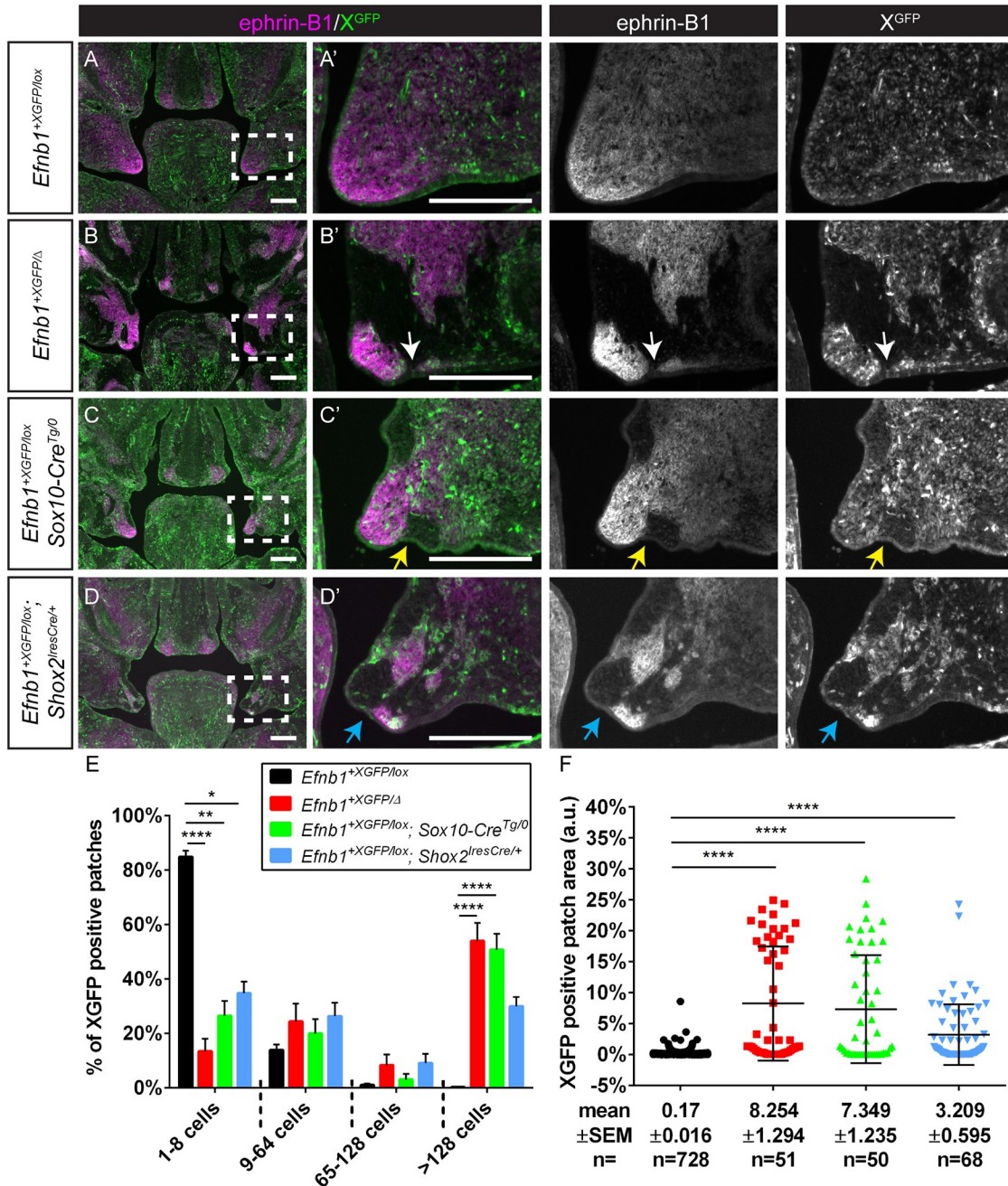

**Fig 3. Craniofacial mesenchyme cell segregation correlates with local dysmorphology in the secondary palate. (A, A')** Immunostaining E13.5 frontal sections for EPHRIN-B1 (magenta) and GFP (green) reveals that EPHRIN-B1 protein is strongly expressed in the anterior-middle palatal shelves. Evenly distributed and intermixed XGFP expressing cells are apparent in control *Efnb1*$^{+XGFP/lox}$ embryos. **(B, B')** Cell segregation is visible in the palatal shelves of *Efnb1*$^{+XGFP/\Delta}$ embryos as large patches of EPHRIN-B1 and GFP expression in these structures. The palatal shelves are also smaller and dysmorphic, with changes in shape occurring at boundaries between EPHRIN-B1 expressing and non-expressing domains (white arrow). **(C, C')** Generation of EPHRIN-B1 mosaicism specifically in neural crest cells using Sox10-Cre results in dramatic cell segregation in *Efnb1*$^{+XGFP/lox}$; *Sox10-Cre*$^{Tg/0}$ palatal shelves, which are smaller and dysmorphic, with regions of dysmorphogenesis correlating with EPHRIN-B1 expression boundaries (yellow arrow). **(D, D')** EPHRIN-B1 mosaicism in Shox2$^{IresCre}$-expressing cells results in cell segregation in *Efnb1*$^{+XGFP/lox}$; *Shox2*$^{IresCre/+}$ palatal shelves. Areas of dysmorphogenesis are visible at the interface between EPHRIN-B1 expression and non-expression domains (blue arrow). **(E)** Distribution of percentage of XGFP-positive patches of various sizes. Column height represents means of the distributions across all sections measured for a given genotype, error bars represent S.E.M., *, P<0.05.; **, P<0.01; ****, P<.0001. **(F)** Patch sizes represented as scatterplots. Horizontal bars represent means, and error bars represent S.E.M. ****, P<.0001. Number of embryos analyzed is presented in S1 Table.

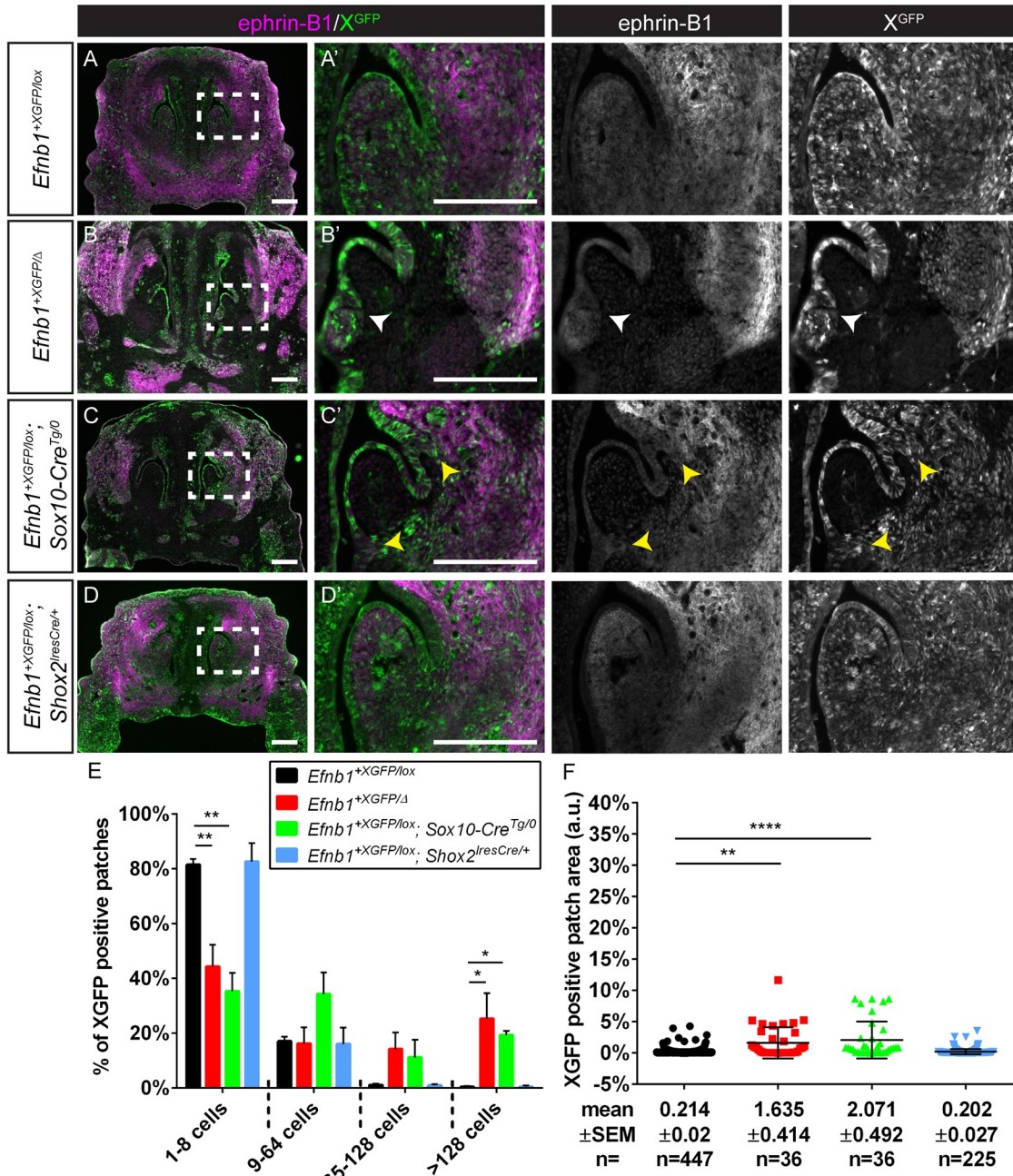

**Fig 4. Craniofacial mesenchyme cell segregation correlates with local dysmorphology in the FNP. (A, A')** Immunostaining of frontal sections of control *Efnb1*[+XGFP/lox] embryos at E13.5 for EPHRIN-B1 (magenta) demonstrates strong expression in the LNP lateral to the nasal concha of the anterior frontonasal process (FNP). XGFP (green)-expressing cells are evenly distributed and intermixed with GFP non-expressing cells. **(B, B')** In *Efnb1*[+XGFP/Δ] embryos with ubiquitous mosaicism for EPHRIN-B1 expression, cell segregation is evident throughout the anterior FNP, and bifurcation of the nasal concha occurs at an aberrant EPHRIN-B1 expression boundary (white arrowhead). **(C, C')** Generation of EPHRIN-B1 mosaicism specifically in neural crest cells in *Efnb1*[+XGFP/lox]; *Sox10-Cre*[Tg/0] embryos results in cell segregation visible throughout the anterior FNP and bifurcation of the nasal concha visible at EPHRIN-B1 expression boundaries (yellow arrowhead). **(D, D')** Restriction of EPHRIN-B1 mosaicism to post-migratory neural crest cells using *Shox2*[IresCre] does not cause cell segregation or dysmorphology in the nasal conchae of the anterior FNP, as *Shox2* is not expressed in this region. *Scale bars, 200 μm.* **(E)** Distribution of percentage of XGFP-positive patches of various sizes. Column height represents means of the distributions across all sections measured for a given genotype, error bars represent S. E.M., *, P<0.05.; **, P<0.01. **(F)** Patch sizes represented as scatterplots. Horizontal bars represent means, and error bars represent S. E.M. **, P<0.01; ****, P<.0001. Number of embryos analyzed is presented in S1 Table.

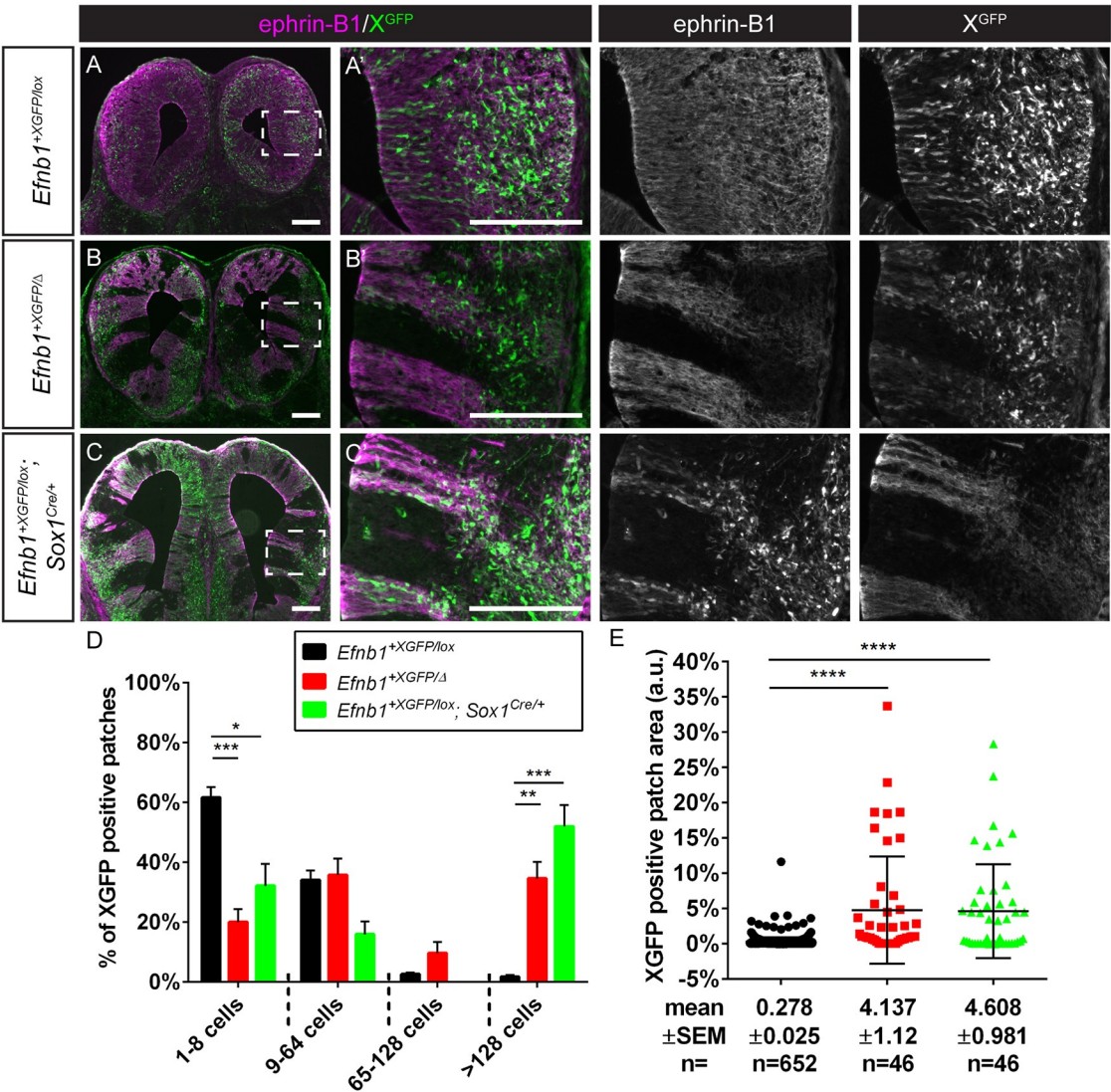

**Fig 5. EPHRIN-B1 mosaicism in neural progenitors produces cell segregation in the brain. (A, A')** Immunostaining of E13.5 coronal sections for EPHRIN-B1 (magenta) and GFP (green) shows high EPHRIN-B1 expression, with an absence of cell segregation as shown by the fine-grained mosaic pattern of XGFP expression. **(B, B')** In $Efnb1^{+XGFP/\Delta}$ embryos with ubiquitous mosaicism for EPHRIN-B1 expression, cell segregation is evident throughout the brain as large patches of EPHRIN-B1 and GFP expression. **(C, C')** Generation of EPHRIN-B1 mosaicism specifically in neural progenitor cells using Sox1$^{Cre}$ results in dramatic segregation throughout the brain of E13.5 $Efnb1^{+XGFP/\Delta}$; $Sox1^{Cre/+}$ embryos, visible as large patches of EPHRIN-B1 and GFP expression. **(D)** Distribution of percentage of XGFP-positive patches of various sizes. Column height represents means of the distributions across all sections measured for a given genotype, error bars represent S.E.M., *, P<0.05.; **P<0.01; ***P<.005 **(E)** Patch sizes represented as scatterplots. Horizontal bars represent means, and error bars represent S.E.M. ****, P<.0001. Number of embryos analyzed is presented in S1 Table.

n = 4) that mirrored what we observed in $Efnb1^{+XGFP/\Delta}$ full heterozygous embryos (Fig 5B, 5D and 5E; n = 5). We therefore quantified the gross facial shape effects of brain-specific EPHRIN-B1 cell segregation in $Efnb1^{+/lox}$; $Sox1^{Cre/+}$ E14.5 embryos with geometric morphometrics. Procrustes ANOVA analysis indicated that $Efnb1$ brain-specific heterozygosity is not a significant contributor to facial shape variation (Table 4). Landmark specific vectors of $Efnb1^{+/lox}$; $Sox1^{Cre/+}$ genotype effects on facial shape are virtually nonexistent (Fig 6A and 6C), and the shape of these specimens overlaps substantially with that of $Efnb1^{wt}$ littermate controls

**Table 4. Significant influence of facial size but not *Efnb1; Sox1-Cre* genotype on facial shape at E14.5 (Procrustes ANOVA).**

|  | Df | SS | MS | Rsq | F | Z | Pr(>F) |
|---|---|---|---|---|---|---|---|
| **Size[a]** | 1 | 0.009 | 0.009 | 0.274 | 8.159 | 3.895 | 0.001* |
| **Genotype[b]** | 1 | 0.001 | 0.001 | 0.021 | 0.617 | -0.039 | 0.464 |
| **Residuals** | 21 | 0.023 | 0.001 |  |  |  |  |
| **Total** | 23 | 0.033 |  |  |  |  |  |

[a]Estimate of the influence of overall size (estimated as centroid size) on facial shape.

[b]Estimate of the influence of genotype (as a factor) on facial shape.

Column abbreviations are the same as defined for Table 1.

(Fig 6E). Each of these observations support the conclusion that neural tissue-specific *Efnb1* heterozygosity does not impact facial shape.

We next quantified the gross facial shape effects of disrupted *Efnb1* expression in NCC-derived tissues. Procrustes ANOVA analysis indicated that *Efnb1*[+/lox]; *Sox10-Cre*[Tg/0] genotype

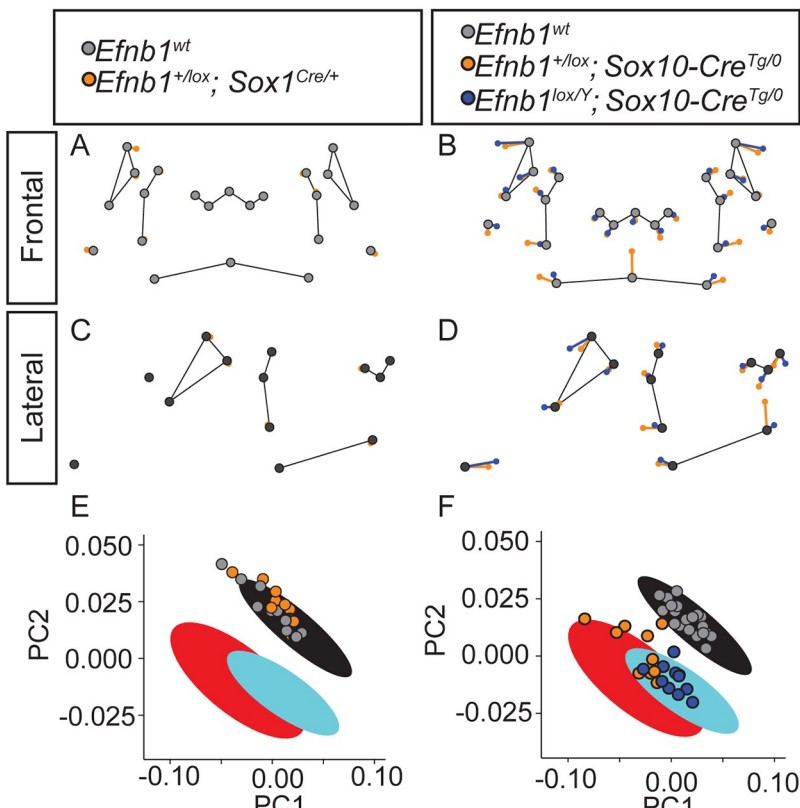

**Fig 6. Disruption of *Efnb1* in NCCs results in face shape changes but disruption in brain does not. (A-D)** Genotype-specific facial shape effects are plotted between predicted E14.5 facial shape landmark positions for *Efnb1*[wt] (grey points) and *Efnb1*[+/lox]; *Sox1*[Cre/+] (orange points) from the **(A)** anterior and **(C)** lateral views and between *Efnb1*[wt] (grey points), *Efnb1*[+/lox]; *Sox10-Cre*[Tg/0] (orange points), and *Efnb1*[lox/Y]; *Sox10-Cre*[Tg/0] (blue points) from the **(B)** anterior and **(D)** lateral views. The lengths of these shape difference vectors are magnified three times to allow for easy comparison of shape effects. Thin black lines are placed for anatomical reference. **(E-F)** Facial shape variation of indicated genotypes is projected along the first two principal components from Fig 1L for direct comparison of *Sox1*[Cre] and *Sox10-Cre* mediated *Efnb1* tissue specific disruption effects with full *Efnb1* genotype effects. The large ovals are the 95% confidence intervals from Fig 1L. Number of embryos analyzed is presented in S1 Table.

had a significant influence on facial shape (Table 5). Landmark-specific vectors of the facial shape effects indicated broadly similar directions of shape change for $Efnb1^{lox/Y}$; $Sox10$-$Cre^{Tg/0}$ hemizygotes and heterozygotes compared with control (Fig 6B and 6D). These include hypertelorism, a relatively inferior rhinarium, and relatively anterior ear. The $Efnb1^{+/lox}$; $Sox10$-$Cre^{Tg/0}$ heterozygotes show increased width of the posterior whisker margins and a higher midline lip when compared to $Efnb1^{lox/Y}$; $Sox10$-$Cre^{Tg/0}$ hemizygotes. As with the comparison of $Efnb1^{+/\Delta}$ and $Efnb1^{\Delta/Y}$ genotypes, the severity of facial shape dysmorphology is less in $Efnb1^{lox/Y}$; $Sox10$-$Cre^{Tg/0}$ males than in $Efnb1^{+/lox}$; $Sox10$-$Cre^{Tg/0}$ heterozygous females (Fig 6F; Table 6). Strong similarities in facial dysmorphology are apparent between embryos with global disruption of $Efnb1$ and those with NCC-specific loss. However, the Procrustes distances between affected mice and wildtype mice are lower for the $Sox10$-$Cre$ crosses (Tables 3 and 6), suggesting a lower severity of facial dysmorphology when cell segregation occurs only in NCC-derived structures. In summary, these morphometric results quantitatively demonstrate that neural-specific disruption of $Efnb1$ has no effect on facial shape in CFNS dysmorphology, while NCC-specific disruption leads to facial shape effects that are similar to but slightly milder than those resulting from global disruption of $Efnb1$ expression.

## Contributions of EPHB receptors to CFNS-like phenotypes and cell segregation

Based on biochemical affinity, EPHB1, EPHB2 and EPHB3 have been proposed to be the principle receptors for EPHRIN-B1 [57]. Though it has been documented that loss of EPHB2 and EPHB3 signaling results in a cleft palate phenotype [58–60], it is currently unknown which receptors are relevant to which CFNS phenotypes, and whether global additive or distinct tissue-specific functions are conferred by each receptor. To determine patterns of expression of these receptors, we performed IF against EPHB2 and EPHB3 and RNA Scope *in situ* hybridization for $Ephb1$ at E13.5. Expression of the EPHB receptors was widespread and diffuse across the secondary palate, brain, and FNP, with detectable expression of each receptor within each of these structures, though with various patterns and apparent degrees of expression (S6 Fig). Examining expression of EPHRIN-B1 by IF in wild-type control and $Ephb1^{-/-}$; $Ephb2^{-/-}$; $Ephb3^{-/-}$ compound mutant embryos revealed no change in the general pattern of expression of EPHRIN-B1 (S7 Fig; **n = 3**), though secondary palatal shelves appeared shorter and less medially extended, thereby truncating the normal EPHRIN-B1 domain of expression. (Fig 7A and 7B'; **n = 3**). In order to illuminate the particular EPH-EPHRIN-B1 interactions that produce CFNS facial dysmorphology, we collected E14.5 embryos harboring all 27 possible genotypic combinations of $Ephb1$, $Ephb2$, and $Ephb3$ null mutant alleles (S1 Table) [59,61,62]. We performed morphometric analysis to identify the phenotypic influence of single and combined EPHB receptor loss. Procrustes ANOVA analysis indicates that genotype of each $Ephb$

**Table 5. Significant influences of facial size and $Efnb1$; $Sox10$-$Cre$ genotype on facial shape at E14.5 (Procrustes ANOVA).**

|  | Df | SS | MS | Rsq | F | Z | Pr(>F) |
|---|---|---|---|---|---|---|---|
| **Size**[a] | 1 | 0.011 | 0.011 | 0.163 | 12.170 | 4.585 | 0.001* |
| **Genotype**[b] | 3 | 0.024 | 0.008 | 0.367 | 9.097 | 6.514 | 0.001* |
| **Residuals** | 35 | 0.031 | 0.001 |  |  |  |  |
| **Total** | 39 | 0.066 |  |  |  |  |  |

[a]Estimate of the influence of overall size (estimated as centroid size) on facial shape.

[b]Estimate of the influence of genotype (as a factor) on facial shape.

Column abbreviations are the same as defined for Table 1.

**Table 6. Procrustes distances[a] of E14.5 facial shapes of *Efnb1* mutant genotypes using tissue-specific Cre alleles.**

|  | Control Male | Control Female | Hemizygous | Heterozygous |
|---|---|---|---|---|
| **Actin-Cre** | 0.03–0.08 (95% CI) | | 0.14* | 0.28* |
| **Sox10-Cre** | 0.07 | 0.07 | 0.16* | 0.24* |
| **Sox1<sup>Cre</sup> ^** | NA | 0.10* | NA | 0.09* |

[a] Higher values represent a greater difference in facial shape, a proxy for severity of dysmorphology.

*indicates a significantly different facial shape than E14.5 *Efnb1*<sup>wt</sup> controls used for comparison to *Efnb1*<sup>+/Δ</sup> and *Efnb1*<sup>Δ/Y</sup>; based on the 95% control confidence intervals produced by bootstrapping.

^ Although both Sox1<sup>Cre</sup> controls and heterozygote facial shapes are significantly different than β-actin-cre controls, they are not significantly different from each other.

receptor gene has significant effects on E14.5 embryo facial shape (Table 7). The proportion of facial shape variation explained by variation in the *Ephb1* null mutation is 1%, while *Ephb2* genotype explains 6% and *Ephb3* genotype explains 10% (Rsq values). Specimens with more null alleles across all three receptors tended to have facial shapes more similar to *Efnb1*<sup>+/Δ</sup> and *Efnb1*<sup>Δ/Y</sup> specimens, but each receptor contributed to facial shape change to a different extent (Fig 7A). For example, specimens that were homozygous null for *Ephb1* often had facial shapes

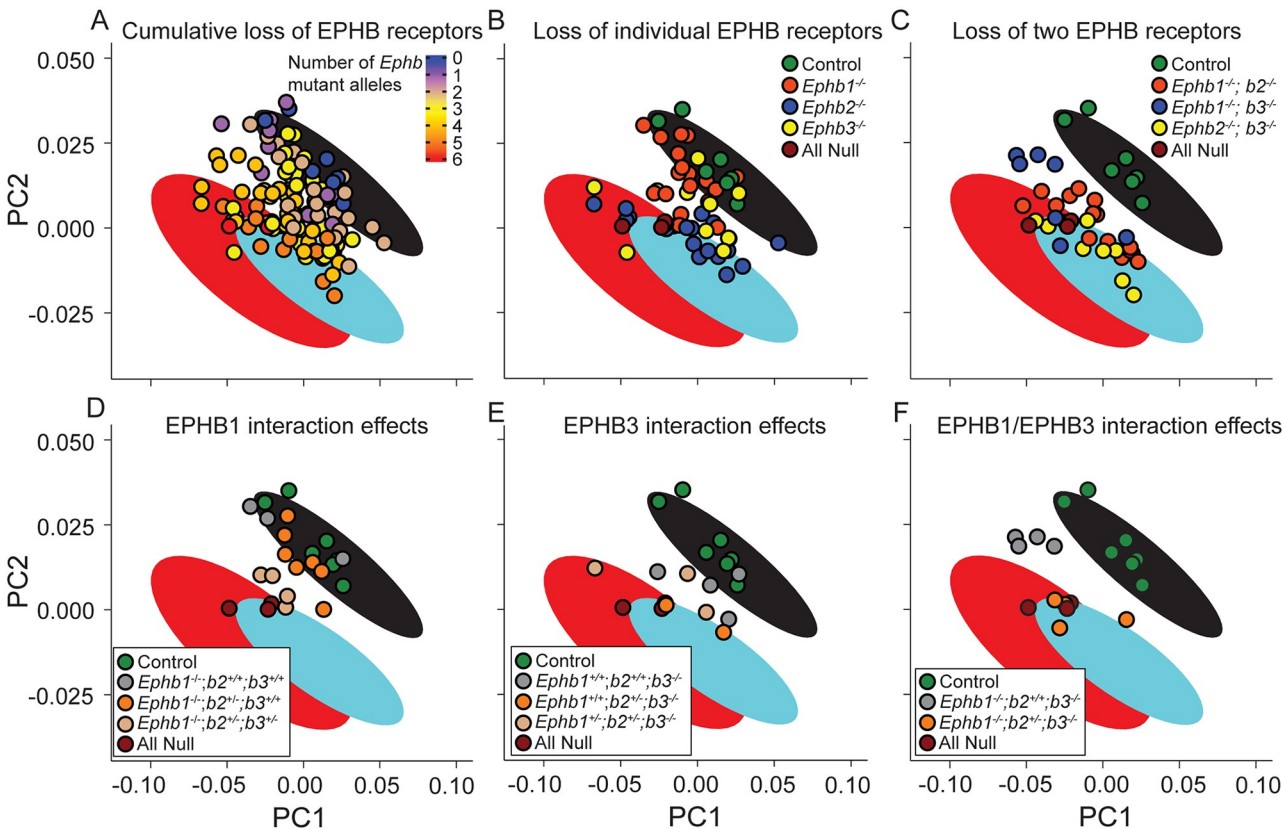

**Fig 7. Distinct EPHB receptors exhibit additive non-equal quantitative effects on face shape.** A sample of all possible *Ephb1*, *Ephb2*, and *Ephb3* null allele genotype combinations displays wide facial variation across the first two principal component axes representing allele facial shape variation (95% CIs from Fig 1L) defined by *Efnb1*<sup>wt</sup> (black ellipses), *Efnb1*<sup>Δ/Y</sup> (cyan ellipses) and *Efnb1*<sup>+/Δ</sup> mutant (red ellipses). **(A)** *Ephb* null series specimens are colored by total number of null alleles. A subset of these specimens that are homozygous null for only one *Ephb* gene **(B)** or two *Ephb* genes **(C)** are plotted alongside EphB *wt* controls and "all null" specimens that are triple *Ephb1*<sup>-/-</sup>; *Ephb2*<sup>-/-</sup>; *Ephb3*<sup>-/-</sup> homozygous mutants. In **(B, C)**, unlisted *Ephb* genotypes include both +/+ and +/-, but not -/- genotypes. Comparisons of specific genotypes illustrate the influence of homozygous and heterozygous genotypes across *Ephb1* **(D)**, *Ephb3* **(E)**, and *Ephb1; b3* homozygous null specimens **(F)**. Number of embryos analyzed is presented in S1 Table.

**Table 7. Significant influences of facial size and *Ephb* receptor genotype on facial shape at E14.5.**

|  | Df | SS | MS | Rsq | F | Z | Pr(>F) |
|---|---|---|---|---|---|---|---|
| **Size[a]** | 1 | 0.049 | 0.049 | 0.247 | 49.583 | 7.546 | 0.001* |
| **EphB1[b]** | 1 | 0.002 | 0.002 | 0.011 | 2.247 | 2.881 | 0.005* |
| **EphB2[b]** | 1 | 0.012 | 0.012 | 0.060 | 12.078 | 6.915 | 0.001* |
| **EphB3[b]** | 1 | 0.019 | 0.019 | 0.098 | 19.589 | 8.411 | 0.001* |
| **Residuals** | 117 | 0.114 | 0.001 |  |  |  |  |
| **Total** | 121 | 0.196 |  |  |  |  |  |

[a]Estimate of the influence of overall size (estimated as centroid size) on facial shape.

[b]Estimate of the additive influence of a specific EphB genotype (as a factor) on facial shape.

Column abbreviations are the same as defined for Table 1.

similar to *Efnb1^{wt}* mice, while specimens that were homozygous null for *Ephb2* usually had facial shapes more similar to *Efnb1^{Δ/Y}* mice (Fig 7B). So, while genotype of each receptor was associated with a significant shape effect, *Ephb1* genotype explained less facial shape variation than *Ephb2* or *Ephb3* genotypes and was associated with less severe phenotypic effects. This finding fits well with the observation that *Ephb1* appeared to be least broadly expressed in craniofacial mesenchyme (S6 Fig).

Interactions between multiple *Ephb* receptor genotypes further explained facial shape variation across this triple null series. For example, some of the variation across specimens that were homozygous null for *Ephb1* resulted from heterozygosity of other receptors. *Ephb1* homozygotes with no other null *Ephb* alleles had facial shapes like *Efnb1^{wt}* mice, indicating weak or no independent impact of *Ephb1*. *Ephb1^{-/-}; Ephb2^{+/-}* embryos also displayed wildtype-like phenotypes (n = 7); however, *Ephb1^{-/-}; Ephb2^{+/-}; Ephb3^{+/-}* embryos exhibited phenotypes more similar to *Efnb1^{Δ/Y}* mutant embryos (Fig 7D; **n = 4**). *Ephb3^{-/-}* null mutants exhibited an intermediate facial phenotype with the severity of dysmorphology increased by *Ephb2* heterozygosity (Fig 7E; **n = 4**). While many specimens that were homozygous null for one receptor gene showed wildtype-like facial shape, most specimens that were homozygous null for two receptor genes displayed more severe dysmorphology (Fig 7C). However, the embryos that were homozygous null for both *Ephb1* and *Ephb3* clustered into two groups along major axes (PCs) of facial shape variation. This separation of specimens was based on whether these specimens were also heterozygous for *Ephb2* (Fig 7F), indicating that having two wild-type copies of *Ephb2* in embryos without EPHB1 or EPHB3 function can lead to a notably milder facial phenotype.

We have previously demonstrated that loss of forward signaling through EPHB2 and EPHB3 resulted in a loss of cell segregation in the neural plate of *Efnb1^{+/Δ}* embryos at E8.5. Because EPHRIN-B1 cell segregation occurring within the post-migratory NCC-derived mesenchyme appears to drive CFNS dysmorphology, we genetically tested which receptors were required for cell segregation in the secondary palate, FNP and telencephalon. We generated compound *Efnb1^{+/Δ}* mutant embryos also harboring loss of function of different combinations of *Ephb1*, *Ephb2* and *Ephb3* alleles and analyzed cell segregation at E13.5 by EPHRIN-B1 immunostaining. Because these embryos do not include XGFP, we first counted the number of cells in EPHRIN-B1 negative patches and plotted the distribution of number of patches with different cell numbers across genotypes. Second, we calculated the average area of EPHRIN-B1 negative patch sizes, with larger patches consistent with more cell segregation having occurred and smaller patches indicating less segregation. Robust segregation with large patches of EPHRIN-B1 positive and negative cells was apparent in the secondary palate and FNP

mesenchyme of $Efnb1^{+/\Delta}$ embryos with most combinations of $Ephb$ genotypes (Fig 8A–8F, 8I and 8J; S8A–S8F, S8I, S8J Fig). Strikingly, $Efnb1^{+/\Delta}; Ephb1^{+/-}; Ephb2^{-/-}; Ephb3^{-/-}$ mutant embryos exhibited reduced segregation in the craniofacial mesenchyme, with significantly smaller EPHRIN-B1 negative patches and more intermixing resulting in an increased number of smaller patches of EPHRIN-B1 expressing and non-expressing cells (Fig 8G, 8I and 8J; S8G, S8I and S8J Fig; **n = 3**). $Efnb1^{+/\Delta}; Ephb1^{-/-}; Ephb2^{-/-}; Ephb3^{-/-}$ embryos exhibited the most dramatic reduction in cell segregation, though regions of EPHRIN-B1 negative cells were still observed to cluster together (Fig 8H, 8I and 8J; S8H, S8I and S8J Fig; **n = 3**). Thus, even complete loss of EPHB1, EPHB2 and EPHB3 was not sufficient to completely abrogate EPHRIN-B1-mediated cell segregation in the palate and FNP, suggesting that additional receptors may contribute to cell segregation in this context. In the telencephalon, a somewhat different priority of receptor requirement was observed. Again, cell segregation was apparent in most $Efnb1^{+/\Delta}; Ephb1-3$ compound mutant embryos, though the extent of intermixing and distribution of patches was different with different receptor combinations (S9 Fig). Notably, EPHB1 seems to play a more important role in cell segregation in the telencephalon, as $Efnb1^{+/\Delta}; Ephb1^{-/-}; Ephb2^{-/-}; Ephb3^{+/-}$ embryos exhibited dramatic loss of cell segregation (S9E, S8I and S8J Fig; **n = 3**) that was similar to that observed in $Efnb1^{+/\Delta}; Ephb1^{-/-}; Ephb2^{-/-}; Ephb3^{-/-}$ embryos (S9H, S9I and S9J Fig; **n = 3**).

## Discussion

From its description as a subgroup of frontonasal dysplasia that affects females more severely than males and the discovery of its X-linked inheritance, CFNS etiology has been mysterious [3,63]. Mouse knockout studies greatly facilitated the identification of *EFNB1* as the responsible gene and implicated the involvement of Eph-ephrin cell segregation [4,5,41,42]. Aberrant EPHRIN-B1-mediated cell segregation, or "cellular interference," is a likely causative mechanism for producing craniofacial and skeletal phenotypes in CFNS patients [37,39,42–44]. It has remained difficult, however, to definitively demonstrate the connection between cell segregation and craniofacial dysmorphogenesis.

Using morphometric analysis in a wide range of mouse genetic models, we have determined the facial changes associated with CFNS pathogenesis and their timing. Significantly wider and shorter faces in *Efnb1* mutant mice were noted as early as E11.5 and increased in severity by E14.5. During this period, which approximately corresponds to weeks 5–8 in human embryonic development, both $Efnb1^{\Delta/Y}$ null hemizygous and $Efnb1^{+/\Delta}$ mosaic heterozygous embryos exhibit changes in facial shape relative to control embryos, but the changes are more pronounced in mosaic heterozygous embryos, presaging the increased severity ultimately seen in heterozygous female CFNS patients. The quantification of phenotypic shape changes in these embryos revealed that dysmorphology analogous to CFNS phenotypes seen in humans with *EFNB1* mutations arose very early during facial morphogenesis, including hypertelorism, midfacial hypoplasia, and higher severity of dysmorphology in females. Specifically, a larger increase in relative width of the midfacial region in the female $Efnb1^{+/\Delta}$ heterozygotes is not matched by the male $Efnb1^{\Delta/Y}$ hemizygotes. In addition, the degree of facial shortening in the females is more extreme, as seen by longer vectors at the ear and nose landmarks. Finally, the female heterozygotes display a much higher point of fusion between the right and left sides of the upper lip. These results indicate that increased midface expansion, arising early in development and not as a consequence of craniosynostosis, underlies more severe phenotypes in female heterozygotes. The strong similarities present in both $Efnb1^{\Delta/Y}$ and $Efnb1^{+/\Delta}$ mutant genotypes indicate that the more severe craniofacial phenotypes noted in female heterozygotes are based in a quantitative extension of dysmorphologies shared with

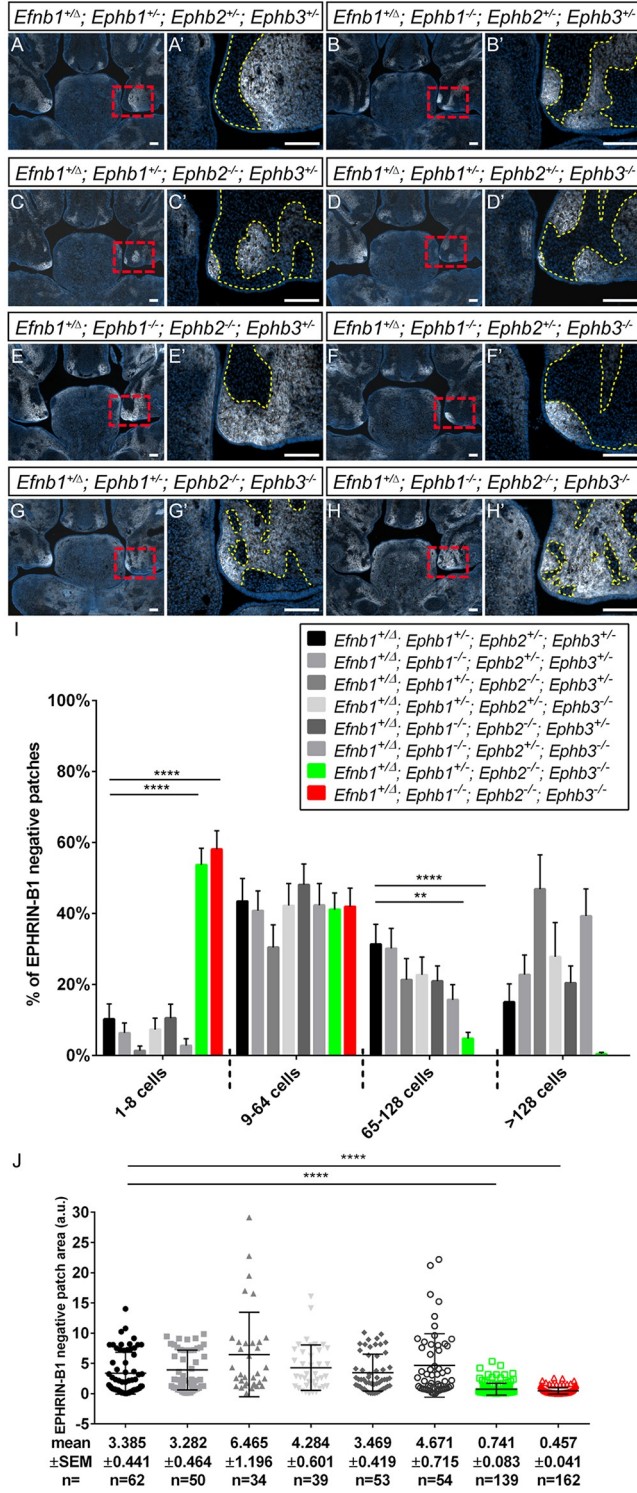

**Fig 8. EPHB2 and EPHB3 receptors mediate cell segregation in secondary palatal shelves.** Secondary palatal shelves of E13.5 embryos harboring compound loss of *Ephb1-3* receptors in combination with *Efnb1*$^{+/Δ}$ heterozygosity with specific genotype combinations shown. Immunostaining for EPHRIN-B1 expression (white) and DAPI (blue) is highlighted with a yellow dashed line at high magnification to demarcate cell segregated patches. **(A-F)** Compound loss of some EphB receptors does not reduce apparent EPHRIN-B1 driven cell segregation, with a relatively small number of large patches of cells observed. **(G, G')** Compound loss of EphB2 and EphB3 receptor resulted in smaller patches, with greater intermingling of EPHRIN-B1 positive and negative cells. **(H, H')** Loss of all known EPHRIN-B1

receptors (EphB1, EphB2, EphB3) also resulted in loss of cell segregation, but with the persistence of small patches of EPHRIN-B1 negative cells. *Scale bars, 100 μm*. **(I)** Distribution of percentage of EPHRIN-B1 negative patches of various sizes. Column height represents means of the distributions across all sections measured for a given genotype, error bars represent S.E.M., **, P<0.01; ****, P<.0001. **(J)** Patch sizes represented as scatterplots. Horizontal bars represent means, and error bars represent S.E.M. ****, P<.0001. Number of embryos analyzed are presented in S1 Table.

male hemizygotes. Given that heterozygotes display cell segregation and hemizygotes do not, it might be expected that $Efnb1^{+/\Delta}$ phenotypes would represent a combination of $Efnb1^{\Delta/Y}$ and qualitatively novel shape effects that are specific to the heterozygotes. However, our results support a fundamentally different situation where hemizygotes and heterozygotes largely exist along a shared quantitative spectrum of facial dysmorphology.

To begin to determine how cell segregation relates to more severe CFNS phenotypes, it is necessary to understand both when (in developmental time) and where (in relevant tissues to CFNS) cell segregation occurs. By generating tissue-specific mosaicism for EPHRIN-B1, we find that in addition to our previously-documented early wave of cell segregation that occurs in the neuroepithelium, cell segregation also occurs independently in the post-migratory NCCs of the craniofacial mesenchyme. Indeed, neural plate-stage cell segregation does not appear to carry through NCC migration, because in $Efnb1^{+/\Delta}$ embryos, E10.5 post-migratory NCC-derived mesenchyme did not exhibit cell segregation. Instead, EPHRIN-B1 mosaicism within NCCs drove robust cell segregation after E11.5 upon the onset of EPHRIN-B1 expression in this tissue, and mosaicism induced later in the palatal shelf mesenchyme was also able to drive cell segregation (Fig 9). These data underscore that there is not one common time-point, or even cell type, for EPHRIN-B1 cell segregation, but rather that EPHRIN-B1 mosaicism can mediate segregation in a wide range of contexts to give rise to the CFNS spectrum of

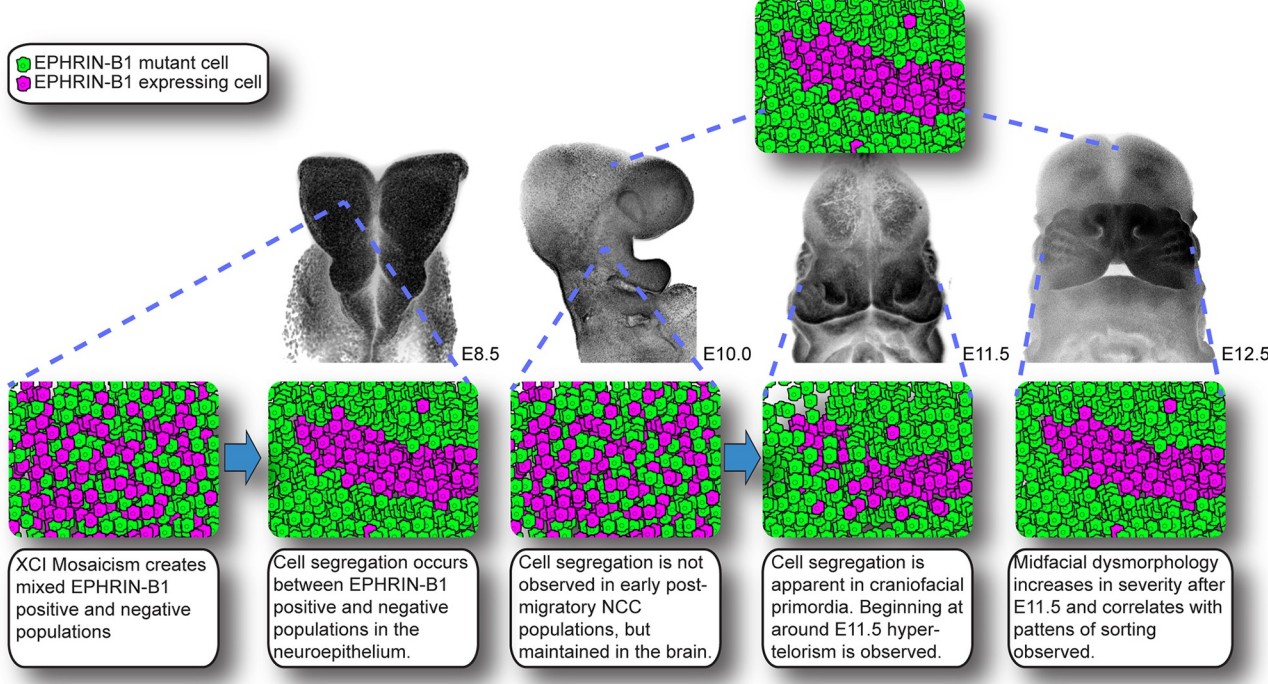

**Fig 9. Model of cell segregation and craniofacial dysmorphology in $Efnb1^{+/-}$ mutant embryos.**

phenotypes. The conserved cellular mechanisms that have such power across dramatically different cell types and developmental time are not entirely known, but likely involve changes in cell adhesion or actomyosin contractility to regulate the strength of cell contacts, or changes in repulsive migratory cell behaviors [44,64–66].

Nevertheless, based on the timing of cell segregation that we document here, together with the timing of quantitative shape changes in *Efnb1* mutant embryos, we infer that CFNS is not caused by defects in NCC migration as previously suggested, but rather reflects a role for *Efnb1* in shaping the craniofacial primordia following migration. Notably, we found that *Efnb1*$^{+/\Delta}$ mutants exhibit changes in tissue shape such as bending, folding and bifurcations in the secondary palate and FNP that correlated with ectopic EPHRIN-B1 expression boundaries. Local dysmorphology as a consequence of XCI-mediated cell segregation cannot explain how both *Efnb1*$^{+/\Delta}$ and *Efnb1*$^{\Delta/Y}$ mutants exhibit qualitatively similar changes in face shape. Whereas cleaved caspase3 staining of control, *Efnb1*$^{\Delta/Y}$, and *Efnb1*$^{+/\Delta}$ E13.5 embryos revealed very few apoptotic cells of the FNP or secondary palate in any of these genotypes (S10 Fig), our previous work demonstrated that cell proliferation is reduced in EPHRIN-B1 mutant regions of the anterior secondary palatal shelves and telencephalon, which may be controlled through the Ras/MAPK signaling pathway [41]. In *Efnb1*$^{\Delta/Y}$ mutants, decreased proliferation may result in reduced tissue growth; in *Efnb1*$^{+/\Delta}$ mutants, this may result in a non-uniform decoupling of growth across these tissues, therefore resulting in more severe local dysmorphology. Additionally, effects in cell position, revealed in *Efnb1*$^{+/\Delta}$ embryos by this unusual X-linked phenomenon, may suggest the existence of previously unappreciated tissue boundaries that exist in the craniofacial mesenchyme that are lost in *Efnb1*$^{\Delta/Y}$ hemizygous males, but ectopically imposed in *Efnb1*$^{+/\Delta}$ embryos. In this way, changes in tissue shape may be the result of effects on cell position in *Efnb1*$^{\Delta/Y}$ mutant embryos. Changes to tissue shape caused by either of the effects described above may also be constrained by the normal pattern of EPHRIN-B1 expression, contributing to the appearance of stereotyped dysmorphology. Further studies will be needed to determine how these aberrant boundaries and/or disruption of boundary maintenance contribute to craniofacial phenotypes in heterozygous females and hemizygous males.

Although segregation occurs dramatically in neural precursor cells at the neural plate and is present in the telencephalon of *Efnb1*$^{+/\Delta}$ embryos later in development, restriction of EPHRIN-B1 mosaicism to neural progenitor cells in *Efnb1*$^{+/lox}$; *Sox1*$^{Cre/+}$ embryos does not result in changes to craniofacial structures or changes to face shape, although segregation in the telencephalon remains equally robust in these embryos. Although previous studies have shown that changes to the structure of the brain can alter the shape of the face [11,12], we demonstrate that this is not the case for the developmental etiology of craniofacial dysmorphology in CFNS. This is somewhat surprising, given 1) the high level of expression of EPHRIN-B1 in the developing telencephalon and 2) dramatic disruptions of neuroepithelium morphogenesis reported in *Efnb1*$^{+/\Delta}$ mouse embryos [53]. Rather, tissue-specific mosaicism in NCC-derived facial tissues leads to facial dysmorphology that is similar in nature to the effects of global mosaicism. There is overlap in the range of facial phenotypes displayed by *Efnb1*$^{+/\Delta}$ and *Efnb1*$^{+/lox}$; *Sox10-Cre*$^{Tg/0}$ embryos along two major axes of facial shape variation. However, the average facial shape of *Efnb1*$^{+/\Delta}$ mice is more different from wildtype facial shape than that of *Efnb1*$^{+/lox}$; *Sox10-Cre*$^{Tg/0}$ mice, which we interpret as greater severity of facial dysmorphology. This difference suggests that NCC-specific *Efnb1* mosaicism does not account for all of the facial dysmorphology noted in *Efnb1*$^{+/\Delta}$ mice. There are multiple possible reasons for this. First, it is possible that mosaicism in other tissues may exacerbate dysmorphology that is primarily driven by NCC-specific mosaicism. Potential interacting tissues include mesoderm-derived cell populations that give rise to cranial base skull bones. It is possible that a reduction in cranial base bone length may also contribute to increased apparent facial shortening [24]. It

is also possible that neural tissue-specific changes may exacerbate facial dysmorphology even if neural tissue-specific changes are not their primary driver.

As a signaling partner for EPHB receptor tyrosine kinases, EPHRIN-B1 has complex signaling mechanisms with multiple possible receptors, as well as proposed receptor-independent functions [33,67,68]. Quantitative analysis of face shape in a triple compound mutant series null for different combinations of *Ephb1*, *Ephb2*, and *Ephb3* provides the first analysis of the particular signaling interactions that are critical for normal face shape development relevant to CFNS. *Ephb1* homozygous null mutation contributes little to facial dysmorphology when compared to the other receptors, which is consistent with low apparent levels of *Ephb1* expression that we observe in the secondary palate and FNP. EPHB2, in particular, appears critical for normal facial development. Although homozygous loss of *Ephb3* led to intermediate dysmorphology, the homozygous loss of *Ephb2* led to dysmorphology similar in nature to that seen in $Efnb1^{\Delta/Y}$ embryos and similar to the dysmorphology noted in embryos with homozygous compound loss of function of all three receptors. $Ephb2^{+/-}$; $Ephb3^{+/-}$ compound mutants exhibited genetic interaction, displaying dysmorphology that was absent in either $Ephb2^{+/-}$ or $Ephb3^{+/-}$ individual mutants. In summary, the range of variation in this sample indicates that the loss of EPHB receptors leads to facial phenotypes like that noted in $Efnb1^{\Delta/Y}$ mice, although *Ephb2* genotype appears to have the most pronounced effect, particularly in combination with *Ephb3*, while *Ephb1* has a minimal effect. Loss of all three EPHB receptors did not recapitulate the severity of the $Efnb1^{+/\Delta}$ phenotypes. This is consistent with the observation that XCI-driven mosaicism followed by cell segregation underlies severity of phenotypes. Complete loss of EPHB receptors does not have a mosaic effect, and maximal EPHRIN-B1-mediated cell segregation in the craniofacial mesenchyme requires receptor expression. Interestingly, though complete loss of EPHB1, EPHB2, and EPHB3 resulted in a dramatic reduction in cell segregation in $Efnb1^{+/\Delta}$; $Ephb1^{-/-}$; $Ephb2^{-/-}$; $Ephb3^{-/-}$ embryos, segregation was not completely abolished, suggesting that additional receptors may play a role. Several EPHA receptors are strongly expressed in the secondary palate mesenchyme, including EPHA4, which was reported to interact with EPHRIN-B1 when overexpressed in Cos7 cells [69,70].

Our improved understanding of the timing and receptor partners involved in cell segregation and craniofacial morphogenesis might ultimately be useful for designing molecular therapies that block Eph/ephrin cell segregation, thus potentially ameliorating more severe CFNS phenotypes. Though we have mainly focused on the relative severity of $Efnb1^{+/\Delta}$ mutant phenotypes, it is important to stress that $Efnb1^{\Delta/Y}$ and *Ephb1; Ephb2; Ephb3* compound mutant mouse embryos exhibit significant craniofacial dysmorphogenesis that includes hypertelorism, frontonasal dysplasia, and cleft secondary palate [8,41,58–60]. Though cleft lip and palate are relatively uncommon in CFNS patients relative to other craniofacial features, a recent genome-wide association study suggested that the *EFNB1* locus may also be relevant to non-syndromic cleft lip with or without cleft palate, which underscores the importance of this pathway in normal development as well as in X-linked CFNS [71].

## Materials and methods

### Ethics statement

All animal experiments were performed in accordance with the protocols of the University of California, San Francisco Institutional Animal Care and Use Committee under approval number AN182040-01. Mice were socially housed under a twelve-hour light-dark cycle with food and water *ad libitum*. If single housing was required for breeding purposes, additional enrichment was provided. When necessary, mice were euthanized by $CO_2$ inhalation followed by cervical dislocation.

## Mouse lines

All alleles used for the experiments herein have been previously described. All mice were back-crossed and maintained on a congenic C57BL/6J genetic background. *Efnb1^{lox}*, MGI: 3039289 [8]; *X^{GFP}*, MGI: 3055027 [47]; *Actin-Cre*, MGI: 2176050 [72]; *Sox10-Cre*, MGI: 3586900 [73]; *Shox2^{IresCre}*, MGI: 5567920 [51]; *Sox1^{Cre}*, MGI: 3807952 [56]; *ROSA26^{mTmG}*, MGI: 3716464 [74]; *Ephb1^-*, MGI: 2677305 [62]; *Ephb2^-*, MGI: 2149765 [61]; *Ephb3^-*, MGI: 2149669 [59]. In all experiments, to generate germline disruption of *Efnb1*, Actin-Cre male mice were crossed to *Efnb1^{lox/lox}* female mice which we refer to as *Efnb1^Δ* throughout the text. For a full descrip-tion of genetic crosses used to generate embryos; strain background, sex, and stage of embryos; and numbers of embryos analyzed, please refer to S1 Table.

## Generation of embryos for analysis of cell segregation

An X-linked beta-actin GFP transgene (XGFP) that demonstrates a fine-grained mosaic pat-tern of GFP expression after random XCI in female embryos [42,47,48] was used to visualize XCI as well as cell segregation in all mosaic embryos. Full EPHRIN-B1 heterozygotes were generated using Actin-Cre mice [72]. *Actin-Cre^{Tg/0}; X^{GFP}/Y* male mice were crossed to *Efn-b1^{lox/lox}* female mice to generate both *Efnb1^{+XGFP/lox}; Actin-Cre^{Tg/0}* and *Efnb1^{+XGFP/lox}* control embryos (referred to in the text and figures as *Efnb1^{+XGFP/Δ}* and *Efnb1^{+XGFP/lox}*, respectively). Embryos mosaic for EPHRIN-B1 expression specifically in the neural crest cell (NCC) lineage were generated using Sox10-Cre mice [73], which were crossed to *Efnb1^{lox/lox}* female mice to generate both *Efnb1^{+XGFP/lox}; Sox10-Cre^{Tg/0}* heterozygous mutant and *Efnb1^{+XGFP/lox}* control embryos. Embryos mosaic for EPHRIN-B1 expression specifically in the palate mesenchyme were generated using *Shox2^{IresCre}* [51]. *Shox2^{IresCre/+}; X^{GFP}/Y* male mice were crossed to *Efn-b1^{lox/lox}* female mice to generate both *Efnb1^{+XGFP/lox}; Shox2^{IresCre/+}* heterozygous mutant and *Efnb1^{+XGFP/lox}* control embryos. Embryos mosaic for EPHRIN-B1 expression in early neural progenitor cells were generated using *Sox1^{Cre}*, which drives recombination in neural plate neuroepithelial cells at E8.5 [56]. *Sox1^{Cre/+}; X^{GFP}/Y* male mice were crossed to *Efnb1^{lox/lox}* female mice to generate both *Efnb1^{+XGFP/lox}; Sox1^{Cre/+}* heterozygous mutant and *Efnb1^{+XGFP/lox}* control embryos. For *Ephb* receptor compound mutants, *Efnb1^{lox/Y}; Ephb1; Ephb2; Ephb3* male mice carrying differing numbers of *Ephb* mutant receptor alleles were crossed to *Ephb1; Ephb2; Ephb3; Actin-Cre^{Tg/0}* female mice carrying differing numbers of *Ephb* mutant alleles to generate *Efnb1^{+/Δ}* embryos with various combinations of *Ephb1-3* mutations (S1 Table).

## Immunofluorescence

Embryos were fixed in 4% PFA in PBS, dehydrated through sucrose, embedded in OCT, and frozen in dry ice/ethanol. 12 μm sections were cut using an HM550 (Thermo Scientific) or a CM1900 (Leica) cryostat. Slides were washed with PBS, blocked in 5% normal donkey serum (Jackson ImmunoResearch) and 0.1% Triton-X-100 in PBS, incubated in primary antibody overnight at 4˚C, washed with PBS, and incubated in secondary antibody at room temperature (for antibody information, please refer to S2 Table). Slides were counterstained in DAPI (Milli-pore) in PBS and coverslips were mounted on slides using Aquamount (Thermo Scientific) for imaging. Images were obtained on an Axio Imager.Z2 upright microscope using an Axio-CamMR3 camera and AxioVision Rel.4.8 software (Zeiss).

## Quantification and statistical analysis of cell segregation

For quantification of cell segregation relating to Figs 2–5, S3 Fig, quantification of cell segrega-tion was performed on cryosections immunostained for EPHRIN-B1 and XGFP and

counterstained with DAPI by two approaches. First, continuous XGFP-expressing regions were selected in FIJI/ImageJ and areas were calculated as a measure of the extent of cell segregation. Second, total numbers of nuclei in XGFP-positive patches were counted using the "analyze particles" function in the secondary palate and FNP, or manually in the brain, for which cell density was too great for automated counting. XGFP-positive regions were binned into patch sizes of 1–8; 9–64; 65–128; >128 nuclei and the number of patches in each bin was divided by the total number of patches to determine the percentage of patches of different size ranges which were then averaged. Quantification of cell segregation in Fig 8, S8 and S9 Figs was performed in the same way, except that EPHRIN-B1 negative regions were instead quantified, as these crosses did not harbor the XGFP transgene. Statistical analysis was performed in GraphPad Prism 6. Assumptions of normality were tested using D'Agostino-Pearson omnibus when sample size allowed. As none of our groups passed this test, or for some experiments, sample sizes were not sufficient to perform it, we performed Kruskal-Wallis tests followed by Dunn's test to determine significance.

## Morphometrics specimen and data acquisition

Embryos were collected at embryonic days E11.5, E12.5, E13.5, and E14.5. Embryos were fixed and stored in a mixture of 4% PFA and 5% glutaraldehyde in PBS. After approximately an hour soaking in Cysto-Conray II (Liebel-Flarsheim Canada), micro-computed tomography (μCT) images of embryo heads were acquired with a Scanco μ35 at the University of Calgary or a Scanco μ40 at Stony Brook University with 45kV/177μA for images of 0.012 mm$^3$ voxel size. All facial landmarks were collected on minimum threshold based ectodermal surfaces (downsampled x2) from the μCT images in Amira (Thermo-Fisher). Because of striking changes in the morphology of the face between E11.5 and E14.5, two different landmark sets were required to quantify facial shape across this period. Previously defined ectodermal landmarks [45], minus those previously identified as problematic (i.e. landmarks 2, 7(24), 10(27), 13(30), 17(34), 18(35), 21(38), 22), were used to quantify facial form of E11.5 embryos. A modified and reduced version of this published landmark set was developed to allow for comparison of ectodermal facial form between E12.5 and E17.5, which we used to quantify facial form of our E12.5, E13.5, and E14.5 embryos (S2 Fig; S3 Table). Landmarks used in morphometric analyses are available as downloadable datasets (S4 & S5 Tables).

## Morphometric analysis of *Efnb1* constitutive mutant embryos

Facial landmarks were collected from hemizygote males (*Efnb1$^{Δ/Y}$*), heterozygote females (*Efnb1$^{+/Δ}$*), and control specimens that were sometimes littermates of affected specimens and sometimes came from separate crosses of *Actin-Cre* and C57BL/6J mice. Separate geometric morphometric analyses were carried out for E11.5 specimens and a combination of E12.5-E14.5 specimens using geomorph [75] in R Statistical Software (R Developmental Core Team, 2008). The procedure is described for the E12.5-E14.5 sample first. Procrustes superimposition was performed on landmarks to align each specimen and remove scale from analysis. Procrustes ANOVA analysis, with permutation-based tests for significance, was used to determine whether size (numeric; centroid size), genotype (factor; *Efnb1$^{+/Δ}$*, *Efnb1$^{Δ/Y}$*, *Efnb1$^{wt}$*), age (numeric; 12.5, 13.5, 14.5) and their interactions have a significant influence on facial shape (α = 0.05). We visualized the effects of *Efnb1$^{+/Δ}$* and *Efnb1$^{Δ/Y}$* genotypes on facial shape by plotting differences between predicted genotype-specific shapes estimated from the Procrustes ANOVA multivariate linear model (assuming E14.5 age and average E14.5 centroid size). Given the strong changes in facial shape that normally occur between E12.5 and 14.5, we completed a multivariate regression of facial shape on centroid size to estimate allometry and used

the rescaled residuals of that regression as "allometry-corrected" coordinates for further analysis. Principal component analyses of coordinate values were completed both before and after "allometry correction" to visualize patterns of specimen clustering along major axes of facial shape covariation within the sample. Procrustes distances between mean control and affected facial shapes were calculated from residual landmark coordinates at each age to determine whether genotypes displayed significantly different facial shapes. Significance was determined by comparing Procrustes distances to 95% age-specific confidence intervals that were estimated with 1000 permutations of distances between two randomly selected control groups of 15 specimens. Geometric morphometric analysis of the E11.5 sample was completed in the same way, except without age as a factor in the Procrustes ANOVA analysis and without allometry correction, because only one age was under analysis. The Procrustes distance values, Procrustes ANOVA output values, and other values are not directly comparable between the E11.5 and the E12.5-E14.5 analyses, because a different set of landmarks undergoing independent Procrustes superimpositions were completed for each age group. However, comparisons of the type of facial shape changes associated with genotype within each age group are valuable to determine if phenotypes are affected similarly in both age groups.

## RNA Scope in situ hybridization

*In situ* hybridization of 12 μm sections was performed using the RNAScope Multiplex Fluorescent Reagent Kit v2 (Advanced Cell Diagnostics, cat# 323100) according to the manufacturer's protocol except that antigen retrieval was bypassed, and the protease step was performed for 5 minutes.

## Facial shape comparison of *Efnb1* tissue-specific and *Ephb* mutant series embryos

E14.5 embryos were collected from crosses of *Sox10-Cre*$^{Tg/0}$ or *Sox1*$^{Cre/+}$ males with *Efnb1*$^{lox/lox}$ females to generate embryos to quantify the effects of tissue specific *Efnb1* loss on facial shape (S1 Table). We intercrossed compound *EphB1; EphB2; EphB3* mutants to generate E14.5 embryos with all possible combinations of *EphB1*, *B2*, and *B3* null allele genotypes to compare the effects of receptor loss with the effects of *Efnb1* ligand loss. Separate Procrustes ANOVA analyses were used to identify significant effects of size (numeric; centroid size) and genotype (factor, *Cre; Efnb1*$^{+/lox}$, *Cre; Efnb1*$^{lox/Y}$, *Efnb1*$^{+/lox}$) for the *Sox1*$^{Cre}$ and *Sox10-Cre* samples. Procrustes ANOVA analysis of the EphB series was completed using the number of null alleles for each EphB receptor as separate numeric factors. To visualize the facial shape effects of these genotypes across E14.5 specimens in relation to full *Efnb1*$^{+/Δ}$ or *Efnb1*$^{Δ/Y}$ genotype effects, each specimen was projected onto principal component axes defined with an E14.5 *Efnb1*$^{+/Δ}$-, *Efnb1*$^{Δ/Y}$-, or *Efnb1*$^{wt}$-specific PCA. The 95% confidence intervals of the facial shape of *Efnb1*$^{+/Δ}$, *Efnb1*$^{Δ/Y}$, and *Efnb1*$^{wt}$ genotypes serve as a standard visual baseline across many of the associated figure panels. Procrustes distances between wildtype specimens and each *Efnb1* mutant genotype were calculated to determine whether tissue-specific expression of *Efnb1* null mutations led to significant facial dysmorphology.

## Supporting information

**S1 Fig. Facial shape effects of genotype (E11.5). (A-F)** Facial landmarks identified on representative *Efnb1*$^{wt}$ **(A-B)**, *Efnb1*$^{Δ/Y}$ **(C-D)**, and *Efnb1*$^{+/Δ}$ **(E-F)** E11.5 specimen surfaces. *Scale bars*, *500 μm* **(G-H)** Common facial shape effects of *Efnb1*$^{/Δ/Y}$ (cyan) and *Efnb1*$^{+/Δ}$ (red) cyan genotypes on facial landmark position, compared to *Efnb1*$^{wt}$ (black) from the anterior **(G)** and lateral **(H)** views. The lengths of these shape difference vectors are magnified three times to

allow for easy comparison. Thin black lines are placed for anatomical reference.
(TIF)

**S2 Fig. Facial landmark definitions (E12.5-E14.5).** Facial landmarks used in morphometric analysis of E12.5-E14.5 samples, based on definitions found in S3 Table, identified on lateral (left) and anterior (right) views of a representative E13.5 wildtype specimen. *Scale bars, 1000 μm.*
(TIF)

**S3 Fig. Craniofacial cell segregation first occurs in the post-migratory neural crest-derived mesenchyme, correlating with the onset of upregulation of EPHRIN-B1. (A, A')** Sox10-Cre drives recombination in the NCC-derived MXP mesenchyme and **(B, B')** frontonasal prominence (FNP) of *Sox10-Cre^{Tg/0}; ROSA26^{mTmG/+}* embryos at E10.5. **(C, C')** *Efnb1^{+XGFP/lox}* control MXP and **(D, D')** FNP demonstrate a fine-grained mosaic pattern of XGFP expression at E10.5. EPHRIN-B1 expression is not strong in the maxillae but has begun to be upregulated in the FNP at this stage. **(E, E')** Likewise, neural crest-specific *Efnb1^{+XGFP/lox}; Sox10-Cre^{Tg/0}* heterozygous embryos demonstrate a fine-grained mosaic pattern of XGFP expression in the maxillary prominences at E10.5, indicating that segregation is not carried through from migratory NCCs. **(F, F')** The FNP of E10.5 *Efnb1^{+XGFP/lox}; Sox10-Cre^{Tg/0}* heterozygous embryos shows a small amount of segregation, visible as patches of GFP expression and non-expression, likely because EPHRIN-B1 has begun to be expressed in the FNP at this stage. **(G, G')** The maxillae of full *Efnb1^{+/Δ}* (recombination mediated by Actin-Cre) are also not segregated at E10.5, but segregation can be seen in the neural tissues of these embryos. **(H, H')** Segregation is visible in the developing LNP and in neural tissues of full EPHRIN-B1 heterozygotes. *Scale bars, 200 μm.* **(I)** Distribution of percentage of XGFP-positive patches of various sizes in the E10.5 maxilla. Column height represents means of the distributions across all sections measured for a given genotype, error bars represent S.E.M. **(J)** Distribution of percentage of XGFP-positive patches of various sizes in the E10.5 FNP. Column height represents means of the distributions across all sections measured for a given genotype, error bars represent S.E.M., *, P<0.05.; **, P<0.01; ***, P<.005; ****, P<.0001. Number of embryos analyzed is presented in S1 Table.
(TIF)

**S4 Fig. Palate-specific EPHRIN-B1 mosaicism results in cell segregation in the anterior palate mesenchyme after E11.5. (A, A')** Shox2^{IresCre} drives minimal recombination in the maxillary prominences of *Shox2^{IresCre/+}; ROSA26^{mTmG/+}* embryos at E11.5. **(B, B')** Most membrane GFP-expressing cells also express neurofilament (2H3) and are likely nerve cells of the maxillary trigeminal ganglion; only a few mesenchymal cells have undergone recombination at this stage (white arrows). **(C, C')** By E12.5, *Shox2^{IresCre/+}; ROSA26^{mTmG/+}* embryos express membrane GFP in the palatal shelf mesenchyme as well as **(D, D')** in the nerve cells of the maxillary trigeminal ganglion. **(E, E')** At E11.5, the maxillae of *Efnb1^{+XGFP/lox}* control and **(F, F')** *Efnb1^{+XGFP/lox}; Shox2^{IresCre/+}* heterozygous embryos are indistinguishable; both genotypes demonstrate a fine-grained mosaic pattern of XGFP expression in the maxillary prominences, indicating that no cell segregation has taken place. **(G, G')** At E12.5, control palatal shelves show a fine-grained mosaic pattern of XGFP expression. **(H, H')** Small patches of EPHRIN-B1/XGFP expressing and non-expressing cells (dashed yellow lines) are visible in the palatal shelves of *Efnb1^{+XGFP/lox}; Shox2^{IresCre/+}* heterozygous embryos at E12.5, demonstrating that post-migratory neural crest cells are also subject to segregation mediated by EPHRIN-B1 mosaicism. *Scale bars, 200 μm.*
(TIF)

**S5 Fig. EPHRIN-B1-mediated cell segregation in the brain does not affect development of craniofacial structures. (A-B')** Recombination of the *ROSA26* locus in two different *Sox1^{Cre/+}; ROSA26^{mTmG/+}* embryos leads to widespread membrane GFP expression throughout the brain at E13.5, but minimal membrane GFP expression in **(C-D')** anterior palatal shelves or **(E-F')** anterior frontonasal prominence (FNP). **(G,G')** Immunofluorescence against EPHRIN-B1(magenta) and XGFP (green) demonstrates that mosaicism in early neural progenitor cells mediated by *Sox1^{Cre}* does not drive segregation in neural crest-derived craniofacial structures such as the anterior palatal shelves or **(H, H')** FNP. EPHRIN-B1 expression (magenta) and craniofacial morphology appear normal in these embryos, indicating that neural progenitor cell segregation is an independent process. *Scale bars, 200 μm.* Number of embryos analyzed is presented in S1 Table.
(TIF)

**S6 Fig. EphB receptor expression in secondary palate, FNP, and brain development.** RNA-Scope *in-situ* hybridization analysis of *Ephb1* expression in the **(A, A')** secondary palate, **(D, D')** FNP, and **(G, G')** brain of E13.5 embryos. **(B-C')** Immunofluorescence staining against EPHB2 and EPHB3 in the secondary palate, **(E-F')** FNP and **(H-I')** telencephalon of E13.5 embryos. *Scale bar, 200 μm.*
(TIF)

**S7 Fig. EPHRIN-B1 expression distribution does not appear altered in EPHB1-3 deficient embryos.** Immunofluorescence staining against EPHRIN-B1 in **(A, A', C, C', E, E')** control and **(B, B', D, D', F, F')** *Ephb1^{-/-}; Ephb2^{-/-}; Ephb3^{-/-}* compound mutant embryos does not reveal overt differences in distribution, though the shortened shape of the secondary palatal shelves in *Ephb1^{-/-}; Ephb2^{-/-}; Ephb3^{-/-}* leads to a reduction in the size of the area usually expressing EPHRIN-B1 in the secondary palate (red arrowheads in A', B') (A-B'). *Scale bar, 200 μm.* Number of embryos analyzed is presented in S1 Table.
(TIF)

**S8 Fig. EphB2 and EphB3 receptors mediate cell segregation in FNP.** Frontonasal processes of E13.5 embryos harboring compound loss of *Ephb1-3* receptor genes in combination with *Efnb1^{+/Δ}* heterozygosity with specific genotype combinations shown. Immunostaining for EPHRIN-B1 expression (white) and DAPI (blue) is highlighted with a yellow dashed line at high magnification to demarcate cell segregated patches. **(A-F)** Compound loss of some EphB receptors does not reduce apparent EPHRIN-B1-driven cell segregation, with a relatively small number of large patches of cells observed. **(G, G')** Compound loss of EphB2 and EphB3 receptor resulted in smaller patches, with greater intermingling of EPHRIN-B1 positive and negative cells. **(H, H')** Loss of all known EPHRIN-B1 receptors (EphB1, EphB2, EphB3) also resulted in loss of cell segregation, but with the persistence of small patches of EPHRIN-B1 negative cells. *Scale bars, 100 μm.* **(I)** Distribution of percentage of EPHRIN-B1 negative patches of various sizes. Column height represents means of the distributions across all sections measured for a given genotype, error bars represent S.E.M., *, $P<0.05$; **$P<0.01$; ***$P<.005$; ****, $P<.0001$. **(J)** Patch sizes represented as scatterplots. Horizontal bars represent means, and error bars represent S.E.M. *, $P<0.05$; **, $P<0.01$; ***, $P<.005$; ****, $P<.0001$. Number of embryos analyzed is presented in S1 Table.
(TIF)

**S9 Fig. EphB receptor combinations mediating cell segregation in the telencephalon.** The telencephalon region of the telencephalon of E13.5 embryos harboring compound loss of *Ephb1-3* receptor genes in combination with *Efnb1^{+/Δ}* heterozygosity with specific genotype combinations shown. Immunostaining for EPHRIN-B1 expression (white) and DAPI (blue) is

highlighted with a yellow dashed line at high magnification to demarcate cell segregated patches. **(A-D)** Cell segregation was robust, but variable in its pattern with haploinsufficiency for various EphB receptors. **(E, E')** Compound loss of EphB1 and EphB2 consistently resulted in a dramatic reduction in cell segregation, whereas **(F, F')** compound loss of EphB1 and EphB3 exhibited no apparent reduction in cell segregation and **(G, G')** compound loss of EphB2 and EphB3 was intermediate. **(H, H')** Complete loss of all three EphB receptors resulted in a dramatic reduction in cell segregation that was similar to compound loss of EphB1 and EphB2. *Scale bars*, *100 μm*. **(I)** Distribution of percentage of EPHRIN-B1 negative patches of various sizes. Column height represents means of the distributions across all sections measured for a given genotype, error bars represent S.E.M., *, $P<0.05$; **, $P<0.01$; ***$P<.005$; ****, $P<.0001$. **(J)** Patch sizes represented as scatterplots. Horizontal bars represent means, and error bars represent S.E.M. **, $P<0.01$; ****, $P<.0001$. Number of embryos analyzed is presented in S1 Table.
(TIF)

**S10 Fig. Apoptosis is rare in the secondary palate mesenchyme and not altered in *Efnb1* mutant embryos.** Immunofluorescence staining against EPHRIN-B1(white) and cleaved caspase 3 (magenta) reveals that little apoptosis is found in the secondary palate mesenchyme in control or *Efnb1* mutant embryos. *Scale bar, 200 μm*. TG, trigeminal ganglia Number of embryos analyzed is presented in S1 Table.
(TIF)

**S1 Table. Table of genetic crosses and embryo numbers (see .xls spreadsheet file, Table S1).**
(XLSX)

**S2 Table. Antibody information for immunofluorescence (IF).**
(DOCX)

**S3 Table. Landmarks for E12.5-E14.5 morphometrics analysis.**
(DOCX)

**S4 Table. Landmark dataset for morphometric analysis of E12.5–14.5 specimens.**
(XLSX)

**S5 Table. Landmark dataset for morphometric analysis of E11.5 specimens.**
(XLSX)

## Acknowledgments

We are grateful for the advice and generous support received from Dr. Benedikt Hallgrímsson during the early phases of this research project. We acknowledge Dr. Francis Smith for collecting landmarks for E11.5 morphometrics comparisons and to Isabel Mormile for collecting some of the μCT images. Thanks go to Ace Lewis and Dr. Camilla Teng for contributing images for use in the model figure. Outstanding technical genotyping support was provided by Fang-Shiuan Leung. We are grateful to our colleagues Dr. Ralph Marcucio, Dr. Licia Selleri, and Dr. Nancy Ann Oberheim for their insightful comments.

## Author Contributions

**Conceptualization:** Terren K. Niethamer, Christopher J. Percival, Jeffrey O. Bush.

**Data curation:** Terren K. Niethamer, Teng Teng, Christopher J. Percival, Jeffrey O. Bush.

**Formal analysis:** Terren K. Niethamer, Teng Teng, Christopher J. Percival, Jeffrey O. Bush.

**Funding acquisition:** Terren K. Niethamer, Jeffrey O. Bush.

**Investigation:** Terren K. Niethamer, Teng Teng, Melanie Franco, Yu Xin Du, Christopher J. Percival, Jeffrey O. Bush.

**Methodology:** Christopher J. Percival, Jeffrey O. Bush.

**Project administration:** Jeffrey O. Bush.

**Supervision:** Jeffrey O. Bush.

**Visualization:** Terren K. Niethamer, Teng Teng, Melanie Franco, Yu Xin Du, Christopher J. Percival, Jeffrey O. Bush.

**Writing – original draft:** Terren K. Niethamer, Teng Teng, Christopher J. Percival, Jeffrey O. Bush.

**Writing – review & editing:** Terren K. Niethamer, Christopher J. Percival, Jeffrey O. Bush.

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
