## [Decision Letter · Decision Letter 0]

26 Aug 2019

Dear Dr Bush,

Thank you very much for submitting your Research Article entitled 'Aberrant cell segregation in craniofacial primordia and the emergence of facial dysmorphology in craniofrontonasal syndrome' to PLOS Genetics. Your manuscript was fully evaluated at the editorial level and by independent peer reviewers. The reviewers appreciated the attention to an important problem, but raised some substantial concerns about the current manuscript. Based on the reviews, we will not be able to accept this version of the manuscript, but we would be willing to review again a much-revised version. We cannot, of course, promise publication at that time.

If you decide to revise the manuscript for further consideration at PLOS Genetics, please aim to resubmit within the next 60 days, unless it will take extra time to address the concerns of the reviewers, in which case we would appreciate an expected resubmission date by email to plosgenetics@plos.org.

[LINK]

We are sorry that we cannot be more positive about your manuscript at this stage. Please do not hesitate to contact us if you have any concerns or questions.

Yours sincerely,

Paul A Trainor

Guest Editor

PLOS Genetics

Gregory Barsh

Editor-in-Chief

PLOS Genetics

Reviewer's Responses to Questions

**Comments to the Authors:**

Reviewer #1: Craniofrontonasal (CFNS) syndrome is an X-linked disorder caused by loss-of-function mutations in EPHRIN-B1. Unlike most X-linked disorders, CFNS is phenotypically more severe in females. Increased severity in females has been attributed to functional mosaicism of Ephrin-B1; however, until now, this mechanism has not been demonstrated in the context of craniofacial development. Ephrin-B1 is a key regulator of cellular boundaries and previous studies by the senior author have shown that ephrin-B1 mosaicism leads to cell segregation. In this study, Niethamer et al. investigate the mechanistic connection between ephrin-B1 mosaicism, aberrant cell segregation, and craniofacial dysmorphology. Using mouse genetics and morphometrics, the authors very convincingly demonstrate that ephrin-B1 mosaicism in post-migratory neural crest cells (NCC) induces ectopic cell segregation in cranial mesenchyme at the onset of dysmorphogenesis in the frontonasal process and palatal shelves. They then employ an allelic series of EphB receptor knockout mice to demonstrate that ectopic cell segregation in the cranial mesenchyme is largely mediated by mosaic ephrinB1 signaling through Eph2 and Eph3 receptors. Overall, this is scientifically rigorous and novel study that makes a significant contribution to the mechanistic understanding of craniofacial dysmorphology in CFNS. While the data support the conclusion that increased severity of ectopic cell segregation correlates with increase severity of craniofacial dysmorphology, this correlation should be strengthened by a quantitative analysis of the cell segregation phenotype. In addition, the introduction and results sections need to be re-crafted, as they are at times hard to follow. Enhancing the manuscript’s clarity will make it more accessible to the wide-readership of PLOS Genetics and highlight these exciting findings.

1. (Tables 1-2 and 4-7) Define Df, SS, MS, etc.

2. (Lines 195-197) Sentence needs to be more clearly stated.

3. (Line 224) It needs to be stated that ephrinB1 expression was detected by IF.

4. (Lines 219-231) The ‘n’ for these experiments should be given here and throughout the main text.

5. Quantitative analysis of the ephrinB1-negative (and/or positive) patches induced by Sox10-Cre mediated deletion at E10.5 vs. E11.5 vs. E13.5 would more strongly support the conclusion that an increase in the severity of cell-segregation is linked to an increase in the severity of the phenotype.

6. (Lines 263-265) Again, quantitative analysis of the ephrinB1-negative (and/or positive) patches would strengthen the correlation between cell segregation and the severity of the phenotype.

7. The ephrinB1 hemizygous males show facial dysmorphology that is similar (albeit, less severe) to the heterozygous females. Cell segregation via X-inactivation cannot account for the males’ phenotype. Therefore, the model that functional mosaicism in cranial mesenchyme induces dysmorphology does not seem to explain the craniofacial phenotype in males. Is it expected that the males’ phenotype results form loss of ephrinB1 function rather than functional mosaicism? This should be discussed in more detail.

8. Figure S5- Text says there is no Sox1-Cre activation in the palatal shelves, but GFP+ cells are seen.

9. (Lines 347-353) Cell segregation differences observed in the quad EphB receptor/ephrinB1+/delta allelic series should be measured in order to strengthen the conclusion that the abnormal cell sorting phenotype is partially rescued by KO of Ephb2 and Ephb3.

10. (Lines 388-389) Hemizygous males may exhibit ectopic cell segregation or loss of normal cell segregation not visible/appreciated by the techniques used here (X-GFP and ephrinB1 expression). For example, loss of ephrinB1 signaling may change Eph-ephrin signaling preferences that result in abnormal boundary establishment or weakening of normal boundaries.

11. Figure 6 shows that the compound EphB receptor KOs exhibit hypertelorism, frontonasal dysplasia, and cleft pallet (phenotypes also induced by ephrinB1 mosaicism). Figure 7 shows that the quad EphB receptor/ephrinB1+/delta allelic series has varying severity in cell segregation depending on the EphB receptors deleted. To more completely understand the significance of the cell segregation phenotype, ephrinB1 IF should be carried out in the compound EphB receptor KOs. If KO of the EphB receptors alone does not induce abnormal cell segregation of ephrinB1+ cells, is it hypothesized that the craniofacial phenotype in the EphB receptor KO mice is instead caused by weakening of an EphB-ephrinB1-mediated boundary (see point 10)?

Reviewer #2: In this manuscript, the authors combined mouse genetic approaches together with quantitative analysis of craniofacial structures during development to investigate the cellular mechanisms of CFNS. They have shown the neural crest-specific contributions of Efnb1 mosaicism to craniofacial dysmorphology. They also examined when Efnb1 regulates cellular position during different stages of craniofacial development and which EphB receptors are involved.

Several points need to be addressed before this study can be considered for publication:

1. The authors suggested that the mosaic loss of Efnb1 in Efnb1 heterozygous mutant mice leads to additional phenotypes not seen in Efnb1 hemizygous male mutant mice, caused by cell segregation. What are the cellular and molecular changes caused by mosaic loss of Efnb1 that lead to the cell segregation? Are there any specifically affected cell adhesion molecule or cell migration changes in Efnb1 heterozygous mutant mice?

2. What happens to the non-Efnb1-expressing and Efnb1-expressing cells after cell segregation during development? Do these cells have different rates of proliferation and/or apoptosis after cell segregation? How do those cellular and molecular changes give rise to the observed changes in tissue shape such as bending, folding and bifurcation in the secondary palate and FNP that correlated with ectopic Efnb1 expression boundaries?

3. RNAseq analysis should be performed to compare the different gene expression profiles of wildtype, Efnb1 heterozygous female mutants, homozygous female mutants and Efnb1 hemizygous male mutants to identify the specific downstream changes caused by cell segregation compared to the complete loss of Efnb1.

4. The authors have shown that compound knockout of EphB receptors dramatically reduce cell segregation in Efnb1+/- mutants. Where are those three EphB receptors expressed? Is there any tissue-specific distribution of each different EphB receptor? Could this also explain why different combinational knockouts of EphB receptors reduce cell segregation differently, apart from the interactions between different EphB receptors?

Minor comments:

Fig. 1, Fig. S1, Fig. S2 need scale bars.

**Have all data underlying the figures and results presented in the manuscript been provided?**

Reviewer #1: Yes

Reviewer #2: Yes

PLOS authors have the option to publish the peer review history of their article (what does this mean?). If published, this will include your full peer review and any attached files.

Reviewer #1: No

Reviewer #2: No

---

## [Decision Letter · Decision Letter 1]

29 Dec 2019

Dear Dr Bush,

We are pleased to inform you that your manuscript entitled "Aberrant cell segregation in craniofacial primordia and the emergence of facial dysmorphology in craniofrontonasal syndrome" has been editorially accepted for publication in PLOS Genetics. Congratulations!

Yours sincerely,

Paul A Trainor

Guest Editor

PLOS Genetics

Gregory Barsh

Editor-in-Chief

PLOS Genetics

Comments from the reviewers (if applicable):

Reviewer's Responses to Questions

**Comments to the Authors:**

Reviewer #1: The authors have addressed all my original concerns and here present a rigorous and interesting study. In reading the revision, I only noted that the section title on Line 151 should instead be "Efnb1 mutant genotype..."

Reviewer #2: Apparently, the authors have another study that will address a couple of the questions i raised in the review. All other concerns have been addressed.

**Have all data underlying the figures and results presented in the manuscript been provided?**

Reviewer #1: Yes

Reviewer #2: Yes

PLOS authors have the option to publish the peer review history of their article (what does this mean?). If published, this will include your full peer review and any attached files.

Reviewer #1: No

Reviewer #2: No

**Data Deposition**

http://datadryad.org/submit?journalID=pgenetics&manu=PGENETICS-D-19-01114R1

**Press Queries**

---

## [Editor Report · Acceptance letter]

19 Feb 2020

PGENETICS-D-19-01114R1 

Aberrant cell segregation in the craniofacial primordium and the emergence of facial dysmorphology in craniofrontonasal syndrome 

Dear Dr Bush, 

We are pleased to inform you that your manuscript entitled "Aberrant cell segregation in the craniofacial primordium and the emergence of facial dysmorphology in craniofrontonasal syndrome" has been formally accepted for publication in PLOS Genetics! Your manuscript is now with our production department and you will be notified of the publication date in due course.

With kind regards,

Kaitlin Butler

PLOS Genetics

On behalf of:
